# Are vision language models robust to classic uncertainty challenges?

**Xi Wang**  *xidulu@gmail.com*
*Department of Computer Science*
*Johns Hopkins University*

**Eric Nalisnick**  *nalisnick@jhu.edu*
*Department of Computer Science*
*Johns Hopkins University*

**Reviewed on OpenReview:** *https: // openreview. net/ forum? id= 4lCSYCNfmo*

## Abstract

Robustness against uncertain and ambiguous inputs is a critical challenge for deep learning models. While recent advancements in large scale vision language models (VLMs, e.g. GPT-4o) might suggest that increasing model and training dataset size would mitigate this issue, our empirical evaluation shows a more complicated picture. In this work, we sanity check whether modern VLMs pass the two most "classic" uncertainty quantification challenges: Anomaly detection and classification under inherently ambiguous conditions. Our results based on non-reasoning VLMs released between late 2024 and early 2025 confirm that newer and larger VLMs tend to exhibit improved robustness compared to earlier models, but still suffer from a tendency to strictly follow instructions, often causing them to hallucinate confident responses even when faced with unclear or anomalous inputs. Remarkably, for natural images such as ImageNet, this limitation can be overcome without pipeline modifications: simply prompting models to abstain from uncertain predictions enables significant reliability gains, achieving near-perfect robustness in several settings. However, for domain-specific tasks such as galaxy morphology classification, a lack of specialized knowledge prevents reliable uncertainty estimation. Finally, we propose a simple mechanism based on caption diversity to reveal a model's internal uncertainty, enabling practitioners to predict when models will successfully abstain without relying on labeled data.

## 1 Introduction

Uncertainty quantification of neural networks is an important problem widely studied by the deep learning community, especially in real-world, safety-critical settings such as self-driving cars (Bojarski et al., 2016) and medical imaging (Esteva et al., 2017). Traditional vision models often struggle with uncertainty quantification (Nixon et al., 2019; Guo et al., 2017; Minderer et al., 2021; Gal & Ghahramani, 2016; Ovadia et al., 2019), largely due to their training regime: These models are typically small-scale and trained on specific, highly curated datasets for single tasks. Consequently, when faced with out-of-distribution (OOD) inputs, they frequently make incorrect yet highly confident predictions, posing significant risks in downstream decision-making systems.

To systematically evaluate the uncertainty quantification ability, two of the most widely adopted workload problems are inputs with corruption and OOD inputs from alternative datasets (see Sec. 2.1 for a more thorough discussion). Under corrupted inputs, such as those found in CIFAR-10C (Hendrycks & Dietterich, 2019), models encounter **covariate shift**, leading to degraded accuracy, in which case a model with good uncertainty quantification should lower its confidence accordingly, rather than maintaining overconfidence despite deteriorating performance, a property known as calibration (Guo et al., 2017). OOD inputs typically

| Model 👉 Task 👇 | ResNet trained on *clean* CIFAR-10 | VLM prompted to classify inputs into CIFAR-10 categories |
|---|---|---|
| CIFAR-10C classification | **Challenge**: Test inputs contain *covariate* shifts
**Failure pattern**: Model makes overconfident predictions despite degraded performance.
Seen at training time: No | **Challenge**: Test inputs are inherently visually ambiguous
**Failure pattern**: VLM fails to recognize ambiguity and responds with incorrect guesses, rather than abstaining
Seen at training time: Unknown but very likely |
| CIFAR-10 v.s. not CIFAR-10 | **Challenge**: Test inputs contain *concept* shifts
**Failure pattern**: Model faithfully classifies non-CIFAR-10 images (OOD) into CIFAR-10 categories.
Seen at training time: No | **Challenge**: Inputs fall outside the semantic scope of CIFAR-10 categories.
**Failure pattern**: VLM fails to detect anomaly inputs and instead generates hallucinated results.
Seen at training time: Unknown but very likely |

Figure 1: **Classic uncertainty quantification tasks revisited in the VLMs era.** Using CIFAR-10 *as an example*, we illustrate how corrupted inputs and inputs from outside CIFAR-10 concepts expose *different* challenges and failure modes in small supervised models vs. large vision language models (VLMs, e.g. GPT4o) prompted to do classification, despite sharing the *same* evaluation data.

refer to inputs from a domain different from the training dataset (e.g. test a model trained on CIFAR-10 with SVHN digits), often having significant **concept shift** (i.e. entirely different label space). The goal of the evaluation is to test whether a model can identify such inputs via flagging them as uncertain rather than forcing misclassifications.

Vision language models (VLMs, Liu et al., 2023; Gao et al., 2023; Liu et al., 2024), such as GPT4o, on the other hand, represent a paradigm shift in visual reasoning. These models leverage massive multimodal datasets and undergo self-supervised pre-training followed by extensive instruction tuning, enabling them to perform diverse tasks in a zero-shot manner, using only the image and a natural language prompt at inference time. Importantly, the *vast* pretraining corpora makes it unclear if the **covariate shift** and **concept shift** challenge underlying the aforementioned two evaluation tasks would still hold for VLM, which raises the question: *Are the traditional evaluation tasks of corruption robustness and OOD detection meaningful for VLMs?*

We argue that these tasks still remain highly relevant to the problem of uncertainty quantification, albeit under a new lens. While VLMs may no longer face clear-cut in-distribution v.s. out-of-distribution boundaries, they still encounter practical challenges in handling **visually ambiguous inputs**, or **anomaly inputs** that fall outside the semantic scope defined by a user prompt. These scenarios frequently arise in real-world applications and can reveal significant reliability gaps. In Fig. 1, we present illustrative examples of these challenges, adapted for CIFAR-10 domain.

Therefore, in this paper, we empirically studied 6 non-reasoning VLMs' behavior on these two problems (Sec. 3.1). We begin by evaluating VLM performance on corrupted ImageNet images, finding that although out-of-the-box performance reveals notable vulnerabilities, larger and newer VLMs exhibit improved robustness (Fig. 2a). Furthermore, we show that prompting the models explicitly to "reject ambiguous inputs" substantially enhances their reliability, enabling the models to abstain from making predictions when appropriate (Fig. 2b). Additionally, we study a classic anomaly detection setting using CIFAR-10 vs. non-CIFAR-10 images, where the goal is to determine whether models can correctly reject inputs that fall outside the specified label space, where we again find that simple prompting is sufficient for models to identify and reject anomalies. Lastly, building on these findings, we extend our analysis to more domain-specific and specialized tasks, such as ECG signal classification and galaxy image recognition, where we find that without sufficient domain expertise, VLMs may show degraded or complete failure at providing reliable uncertainty quantification.

Additionally, we observe that VLM's uncertain level about an input image (Sec. 3.2) can be revealed by prompting the VLM to generate multiple captions for the input image under random sampling decoding, where VLMs tend to generate more diverse captions for visually ambiguous samples, which is more likely to be abstained, and vice versa. This insight allows us to predict the model's ability to successfully perform classification with rejection *without relying on labeled ground truth*.

| Prompt | `Classify the provided image into the following ten categories ...` `Notice that if you find an image ambiguous and cannot confidently classify it, return "unknown" as the label.` |
|---|---|

| Image | Model | Llama 3.2 11B (Sep 2024) | Qwen 2 7B (Sep 2024) | Qwen 2.5 3B (Feb 2025) | Qwen 2.5 7B (Feb 2025) | Qwen 2.5 72B (Feb 2025) | GPT4o-mini |
|---|---|---|---|---|---|---|---|
| | Model output | Dog | Person | Automobile | None of the above | None of the provided classes | None of the given classes apply |
| | | unknown | unknown | unknown | unknown | unknown | unknown |

(a) VLMs are prompted to classify a noisy cat ImageNet-C image into CIFAR-10 categories. Black `monospaced text` shows the standard prompt and the corresponding output. Red text shows the appended explicit rejection prompt and the resulting output. Older and smaller models (e.g. Llama and Qwen 2.5 3B) hallucinate labels without the rejection prompt, whereas larger models often reject uncertain inputs even without explicit instructions.

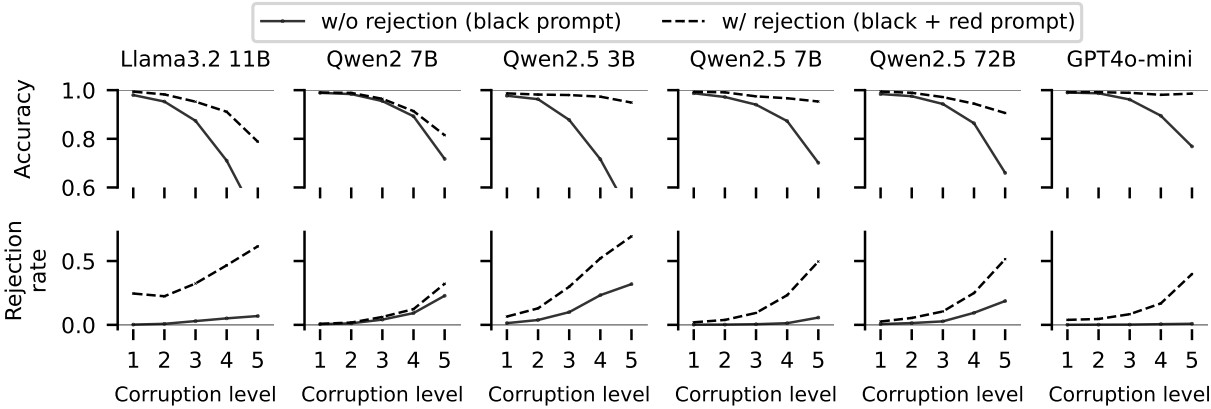

(b) We prompt different VLMs (columns) to classify the same 1,000 samples subset of Gaussian noise-corrupted ImageNet into CIFAR-10 categories, under various corruption intensities (x-axis). Under standard prompt (black part in top of 2a only), the accuracy (among all samples the models output a proper label) decreases as corruption intensifies (solid lines). When models are explicitly prompted to reject ambiguous inputs (black and red parts), the accuracy for classified samples becomes significantly higher (dashed lines). The bottom row shows the corresponding rejection rates for each model: The models reject more inputs as corruption levels increase.

Figure 2: **VLMs show degraded performance under corrupted inputs, allowing rejection helps maintain reliability.** *Top row* demonstrates VLMs' outputs for a selected sample with or without a rejection prompt appended. *Bottom row* shows the classification accuracy under standard prompt without (solid line) vs. with (dashed line) rejection instruction prompt appended.

Over the last decade, much research effort has been spent characterizing the robustness properties of deep neural networks. Central to this research was the development of corruption benchmarks, such as CIFAR-10C, which contaminates the original CIFAR-10 images with various kinds of noise. This was shown to be an enduring benchmark, with SOTA accuracy for a traditional classifier hovering around 85% for CIFAR-10C (Wang et al., 2020)[1]. In this work, we aim to answer the historical question: are these corruption benchmarks like CIFAR-10C, as well as "OOD detection" benchmarks, now "solved"[2] for modern VLMs? And if so, with what generation of VLM did they become "easy"? We find that VLMs have solved this benchmark only relatively recently: models released in 2025 are rather robust, whereas earlier models, such as Llama 3.2, still can easily fail without a proper prompt, despite presumably having seen these datasets during training. We believe our study demonstrates an important lesson: *Despite the clear superiority of modern, large-model AI systems, do not assume they solve older benchmarks.*

---

[1]This result is for methods that do not see noise / corruption during training, only clean images.
[2]When we say "solved", we mean robust by default or with minor modifications.

| Image | Prompt |
|---|---|
| 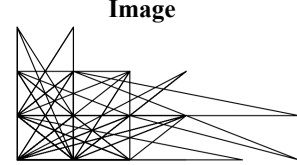 | Your task is to classify the given ECG signal into the normal case and the abnormal cases. Note that if the signal does NOT look like ECG signal or cannot be classified into either of the categories, return ``unknown" as the answer. |

| | |
|---|---|
| Reasoning: The given ECG signal appears to have a regular rhythm with no significant deviations from the expected pattern of a normal ECG. The waves (P, QRS, T) are present and consistent with each other, indicating that there are no signs of arrhythmia or other abnormalities.
**Answer: normal** | Reasoning: The given image appears to be a geometric figure with multiple lines intersecting at various points. However, it does not resemble any standard ECG (Electrocardiogram) signal pattern. An ECG signal typically shows a series of waves representing the electrical activity of the heart over time, which is not present in this image.
**Answer: unknown** |

Figure 3: **Enabling the rejection option allows VLM to pick out anomaly inputs, preventing hallucination.** When prompted to classify a random line image into normal v.s. abnormal ECG signal, certain VLM (Qwen 2.5 3B) would generate hallucinated results (left), potentially caused by its tendency towards strictly following instructions (Sharma et al., 2023). However, when the prompt explicitly permits rejecting non-ECG input (additional red texts in the prompt), the same model correctly identifies the input as anomalous and responds with `Unknown` (right).

## 2 Background and related work

### 2.1 Improving models' robustness against uncertain inputs

Neural networks often struggle with input uncertainty due to their training methodology. When trained on carefully curated datasets with minimal ambiguity, models develop a tendency to produce high-confidence predictions for all inputs, regardless of their clarity or relevance. This behavior creates significant reliability concerns when deploying these models in real-world scenarios where inputs may be ambiguous, corrupted, or entirely outside the scope of the model's expertise. To systematically evaluate models' behavior under uncertain inputs, the deep learning community has identified two primary testing paradigms:

- **Distribution / covariate shift** occurs when a model encounters inputs from a different distribution than its training data, while maintaining the same label space. Examples include synthetic corruptions or stylistic variations (ImageNet-C/P, Hendrycks & Dietterich, 2019), or alternative data collection pipelines (ImageNet-V2, Recht et al., 2019). In these scenarios, models often show degraded accuracy, as these inputs are outside their capacity, while maintaining high confidence predictions. Such a dangerous combination will provide misleading signals for downstream decision-making, whereas ideally, a model should elicit some signals indicating its "unsureness" and express confidence proportional to its likelihood of correctness (Guo et al., 2017).
- **Concept shift (OOD/anomaly detection)** typically considers anomaly inputs coming from entirely different datasets with non-overlapping label spaces than the training distribution. For instance, a digit classifier might encounter images of animals or vehicles. In such cases, the desired behavior is for the model to recognize the mismatch and refuse classification entirely, rather than confidently assigning inputs to irrelevant categories. This capability, often called OOD/anomaly detection, is crucial for safe deployment in open-world environments (Nalisnick et al., 2018; 2019b).

Standard deep neural networks without specialized training typically fail at both tasks. Researchers have developed numerous approaches to address these limitations. The central idea of many of these methods lies around tweaking the training loop in a way such that the model becomes capable of eliciting uncertainty information through the statistics (e.g. entropy) of the predictive distribution, such as Bayesian neural networks (MacKay, 1992; Neal, 2012; Graves, 2011; Blundell et al., 2015; Gal & Ghahramani, 2016; Maddox et al., 2019; Aitchison, 2020a;b; Daxberger et al., 2021a;b; Nalisnick et al., 2019a; Izmailov et al., 2021; Wenzel et al., 2020), deep ensemble (Lakshminarayanan et al., 2017; Abe et al., 2022; D'Angelo & Fortuin, 2021), outlier exposure (Hendrycks et al., 2018), with which practitioners can be aware of when distribution

shift / anomaly inputs shows up. There are also methods aiming at improving the models' generalization to certain distribution shift, such as test time adaptation (Wang et al., 2020; Wang & Aitchison, 2023; Schirmer et al., 2024; Nado et al., 2020; Schneider et al., 2020) for improving accuracy against corrupted inputs, and to anomaly inputs, such as open set classification (Geng et al., 2020) or meta learning (Hospedales et al., 2021).

Initially, one might think that such tasks pose *little* challenge for VLMs, based on two arguments:

1. **No clear OOD boundary**: Traditional models (e.g. ResNet trained on CIFAR-10) face clear distribution boundaries; in contrast, VLMs trained on extensive and diverse multimodal datasets inherently blur these distinctions.
2. **Scaling improves uncertainty quantification**: Prior observations suggest that *simply* scaling models and datasets can significantly mitigate uncertainty quantification problems. Indeed, prior studies highlight improvements in OOD detection (Fort et al., 2021) and calibration under distribution shift (Minderer et al., 2021) with larger-scale models.

However, upon closer investigation, these arguments do not fully hold:

1. Indeed the distribution shift challenges represented by some of the dataset, such as ImageNet-V2/P may no longer be applicable for VLMs, there still exists certain real-world uncertainties, such as *corruptions caused by sensor malfunctions* or *accidental user uploads*, that can be simulated by datasets like ImageNet-C [3] and *anomaly detection settings*, that remain practically significant.
2. State-of-the-art *large-scale* VLMs may exhibit non-negligible issues due to their instruction-following nature (Sharma et al., 2023), when no special prompt is included, they frequently resort to incorrect or random predictions when faced with uncertain or anomalous inputs

Importantly, conventional methods for improving robustness, such as Bayesian neural networks or ensemble techniques, are impractical for VLMs due to the enormous computational overhead required for fine-tuning. Similarly, test-time interventions like test time adaptation or temperature scaling are difficult to implement given the black-box nature of many of these models.

Encouragingly, we discovered that the implicit uncertainty quantification capabilities embedded in VLMs, derived from their instruction-following behavior, offer a simple yet effective solution. Directly prompting VLMs to reject uncertain inputs rather than forcing classification significantly enhances their robustness without necessitating architectural changes or specialized training. This implicit form of uncertainty handling capitalizes on the model's inherent ability to condition its behavior on input instructions, thus providing practitioners with a straightforward approach to improving reliability in practical deployments.

## 2.2 Uncertainty quantification in large language models

A large body of work investigates uncertainty quantification in large language models (LLMs), in the absence of multimodal inputs. One line of research explores the confidence score elicited by the model itself, where models are prompted to express their own certainty about their outputs (Xiong et al., 2023). Another class of methods leverages sampling-based approaches, such as measuring the number of semantic clusters among independently sampled completions to infer semantic consistency and model confidence (Kuhn et al., 2023; Nikitin et al., 2024). Lastly, statistics of the logits themselves also provide informative signals for uncertainty quantification (Kadavath et al., 2022). We hypothesize that such techniques are transferable to multimodal tasks, including those we examine. However, as shown in our experiments, prompting the model with an additional line of instruction that enables rejection is sufficient for the task we considered. Nevertheless, the aforementioned approaches could augment this with more explicit and numerically-scored uncertainty estimates. To be more specific, compared with the aforementioned approaches, our suggested prompting approach works with pure black-box LLM access, requiring access to logits, nor does it require multiple independent samples from the model outputs. However, the cost of simplicity is the lack of controllability: Prompting does not allow us to adjust the level of conservatism for rejection, as we do not have access to a numeric uncertainty score to threshold on.

---

[3]Corrupted inputs are typically seen as a distribution shift from clean training distribution, but the "noise inherent in the observations" (definition of aleatoric uncertainty, Kendall & Gal, 2017) also introduces ambiguous visual cues, making the corrupted inputs hard to recognize visually.

There also exists a line of work that "Bayesify" a pre-trained foundation model through applying Bayesian inference on LoRA adapter during fine-tuning time to equip the fine-tuned model with better uncertainty quantification ability, via e.g., ensemble (Wang et al., 2023), Laplace approximation (Yang et al., 2023a), or variational inference (Wang et al., 2024c). Certainly, these approaches can be applied on VLMs to construct a "Bayesian VLM", however, the scope of these approaches is limited to fine-tuning on multiple-choice questions, and the applicability of free-form generation tasks on pre-trained models remains unclear, in contrast to the prompting approach.

### 2.3 Uncertainty quantification in vision language models

Several recent works have explored uncertainty quantification in the context of vision-language *embedding models* such as CLIP (Radford et al., 2021) and DINO (Caron et al., 2021), focusing on the quality and uncertainty of the learned representations (Miao et al., 2024; Fillioux et al., 2024; Cui et al., 2024). However, these embedding models are not the focus of our work, we are studying LlaVA (Liu et al., 2023) style full VLM that can follow natural language instructions.

In the domain of full VLMs, a few studies examine their uncertainty estimation capabilities. For instance, Kostumov et al. (2024) and Groot & Valdenegro-Toro (2024) investigate the calibration of VLMs on *standard* Visual Question Answering (VQA) benchmarks, paralleling the calibration protocols established in the LLM literature discussed in the previous section. However, these works focus on standard benchmarks where there is limited inherent uncertainty in the inputs. Recent work Miyai et al. (2024a;b) introduce the concept of Unsolvable Problem Detection (UPD), where the goal is to determine whether a given image-question pair is unanswerable. Their Intrinsically Visual Question Detection (IVQD) task is most similar to our anomaly detection setup, in which a mismatch between the question and image indicates an uncertain or ambiguous case. However, our evaluation differs in construction: while they rely on manually annotated datasets, we automatically generate our test data in a manner aligned with classic out-of-distribution (OOD) detection literatures, enabling more scalable evaluation. Moreover, their work focuses on uncertainty arising from the interplay between visual and textual modalities, our experiments additionally studies the ambiguity and uncertainty from images alone. Zhang et al. (2024) considers the interaction of VLMs with corrupted image, similar to our work, however their goal is to perform *hallucination detection* by checking VLMs output variability under various corrupted versions of the same input, while our work aims at evaluating VLMs' *robustness* against corrupted inputs.

## 3 Methods

### 3.1 Evaluation of VLM's implicit uncertainty quantification ability

Our main contribution centers on assessing VLMs' ability to recognize and express uncertainty through *rejecting* problematic inputs, when performing image classification tasks through natural language prompting. Drawing inspiration from the literature on out-of-distribution robustness (Sec. 2.1), we design two complementary evaluation tasks:

**Anomaly detection with prompting**  This task evaluates a model's ability to identify inputs that fall outside the provided category definitions. We prompt the model to reject inputs that do not belong to any of the specified categories. For *evaluation*, we treat samples inside the given categories as the negative class and other samples as the positive class, employing standard binary classification metrics: precision (what fraction of rejected inputs are actual anomalies) and recall (what fraction of all anomalies are successfully rejected), to measure performance.

**Classification with rejection against ambiguous inputs with prompting**  In this task, we present models with inputs that exhibit varying degrees of inherent ambiguity, making them potentially classifiable into multiple categories. We then explicitly prompt the model to reject inputs it finds difficult to classify into a single category. Our hypothesis is that as ambiguity increases, models without a rejection option will resort to random guessing, leading to increased error rates. Conversely, models with a rejection option can abstain from classification rather than make low-confidence predictions. To *evaluate* performance, we measure accuracy on the subset of non-rejected samples, with the ideal behavior being perfect accuracy after rejection, demonstrating the model's ability to recognize when it might make errors.

| Image | Caption |
|---|---|
| | The image shows a red car that has been severely damaged, likely in an accident… |
| | The image shows a red car with significant front-end damage, likely after a collision or accident… |
| **Clear** car | The image shows a damaged red car with visible dents and a broken windshield… |
| | The image appears to show a _street scene with a vintage-style tram_ or trolley in the foreground… |
| | The image depicts a _colorful and vibrant indoor setting with numerous_ people gathered in what appears to be a festive or celebratory atmosphere… |
| **Ambiguous** car | The image appears to be a _pixelated or heavily distorted photograph_ of a street scene… |

(a) ImageNet-C

| Image | Caption |
|---|---|
| | The image shows a spiral galaxy with a distinct bar structure running through its center… |
| | The image shows a spiral galaxy with a prominent central bulge and a distinct spiral structure… |
| **Clear** edge on disk | The image depicts a spiral galaxy with a prominent, well-defined bar structure running through its center. |
| | The image depicts a spiral galaxy with a bright, diffuse central core and a visible disk structure. |
| | The image depicts a spiral galaxy with a prominent central bulge and a disk structure.. |
| **Ambiguous** unbarred spiral | The image shows a spiral galaxy with a prominent central bulge and a spiral arm structure extending outward. |

(b) Galaxy Zoo

Figure 4: **VLMs generate diverse captions for ambiguous images.** We prompt Qwen2.5 7B to "generate a description" given an input image under different random seeds. Clean image from ImageNet receives consistent captions while its corrupted version having a diverse set of captions (top left v.s. bottom left). For the galaxy image where annotators show significant disagreement on whether there exist spiral arms (bottom right), VLMs fail to have diversity in the caption, indicating that the model does not understand the ambiguity, likely due to limited domain knowledge.

While these evaluation paradigms root in the study of out-of-distribution robustness, a concept not applicable to VLMs, they still represent real-world challenges that VLMs must overcome in practical applications. Additionally, unlike previous approaches that required model modifications or specialized training, our method for uncertainty quantification leverages the inherent capabilities of VLMs through just prompting. It is also worth noting that our method for quantification uncertainty is _implicit_ in that our model only has a binary option: Classify the inputs or reject, instead of eliciting a continuous score for uncertainty level.

## 3.2 Caption diversity for understanding the underlying mechanism of rejection

When adopting uncertainty quantification methods for non-black-box models, we typically have access to a continuous score that reflects _how uncertain the model finds a given input to be._ Common examples include the degree of disagreement among ensemble components (Abe et al., 2022), the typicality of a test input relative to the training distribution (Nalisnick et al., 2019b), or the distance of test inputs from the training data measured in a kernel space (Liu et al., 2020; Immer et al., 2021). In contrast, when VLMs are prompted to reject ambiguous inputs, the only available feedback is binary: whether the input was classified or rejected. This raises a critical question, can we have an "ambiguity score" that tells us _how ambiguous an image is for a particular VLM_, such that the higher score an input receives, the more likely it will be rejected by a VLM.

We propose using caption diversity for diagnosing VLM's understanding of input ambiguity. Intuitively, when a model encounters an ambiguous image, one permitting multiple plausible interpretations, it produces a more diverse set of captions across independent generations (under random sampling decoding). Conversely, clear and unambiguous images yield consistent descriptions. To quantify this, we compute a **caption diversity score** by first embedding all generated captions with a sentence transformer model (`all-mpnet-base-v2` from Reimers & Gurevych, 2019), then calculating one minus the averaged pairwise cosine similarity among the embeddings. Our experiments indeed support this hypothesis (Fig. 4a and Fig. 5): As input images become more ambiguous due to higher corruption level, the overall caption diversity increases. Additionally, the images model chooses to reject consistently exhibits higher caption diversity than the classified ones.

Beyond serving as an analytical tool, caption diversity also offers a practical mechanism for predicting rejection behavior without labeled data. By examining the relationship between diversity scores and input ambiguity, practitioners can assess whether a model is likely to abstain from unreliable predictions. Notably, our experiments reveal that for specialized domains requiring expert knowledge, such as galaxy morphology,

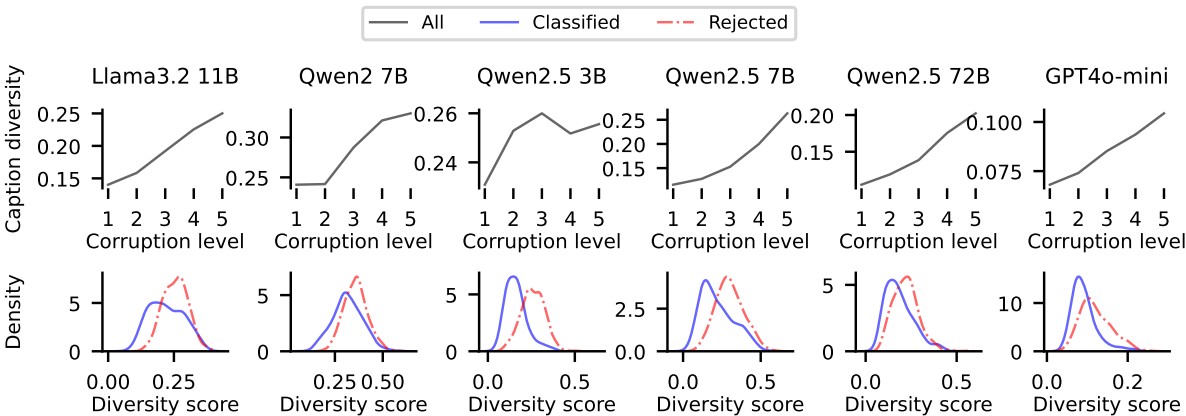

Figure 5: **Caption diversity reflects model uncertainty under ambiguous inputs.** We empirically verified the hypothesis from Sec. 3.2. *Top:* As corruption increases, caption diversity rises across all models, indicating greater uncertainty. *Bottom:* Rejected samples exhibit higher caption diversity than classified ones, suggesting that diversity of independently-generated captions correlates with models' internal uncertainty level for an input and the tendency for rejecting it when prompted.

models generate low-diversity captions even for ambiguous inputs, failing to recognize their own uncertainty (Fig. 4b) and leading to ineffective rejection (Fig. 6).

**Scope and limitation of caption diversity** Lastly, it is worth noting that we do not view caption diversity as a generic uncertainty estimator in that it only captures the uncertainty in the inputs, as it is not conditioned on the query. Therefore, it does not reflect query-dependent uncertainty, e.g, the query involves non-existent components in the image. Additionally, visually ambiguous images are *not the only* type of input that could trigger diverse captions. When the inputs contain a significant amount of information, e.g. an image of a scene with hundreds of people, the model may also output diverse captions under random decoding where different random samples capture different perspectives/details of the inputs (Chan et al., 2023).

## 4 Experiments

In this section, we empirically evaluate the two tasks proposed in Sec. 3.1 on the following families of VLMs: GPT4o-mini (Achiam et al., 2023), Llama 3.2 (Dubey et al., 2024), and Qwen 2/2.5 (Wang et al., 2024a; Bai et al., 2025). GPT4o-mini has a version of `2024-07-18`, Llama 3.2 is released in September 2024, Qwen 2 and 2.5 are released in September 2024 and February 2025 respectively. For the Qwen2.5 72B, we use the official released AWQ quantized version.

For all experiments, we treat all models as black boxes, where we compute the metrics by looking at the output string without using the information from the logits. Unless explicitly stated, we use deterministic sampling for querying the VLMs. For experiments that use multiple generations from random decoding, we use a temperature of 0.6, a top-P of 0.95, and top-K of 50 for all models. We conduct all experiments on an internal cluster of Nvidia H100s and A100s.

The complete prompt used are shown in Appendix. A. We conduct ablation study over the prompting style (Appendix. E): Broadly, we find that the ability of rejection is sensitive to prompting style for Llama and Qwen2, but the rest models are less sensitive to the prompting mode.

### 4.1 Anomaly detection

We begin by examining anomaly detection tasks across two datasets:

**CIFAR-10 v.s. Not CIFAR-10**: We selected images from ImageNet (Deng et al., 2009) with concepts overlapping with CIFAR-10 categories as the target classification samples (detailed class mapping shown in Appendix. B), then considered images outside these concepts as anomalies. We prompted the models

| Model | CIFAR-10 v.s. Not CIFAR-10 | | | ECG v.s. Not ECG | | |
|---|---|---|---|---|---|---|
| | **Precision ↑** | **Recall ↑** | **F1 ↑** | **Precision ↑** | **Recall ↑** | **F1 ↑** |
| Llama 3.2 11B | 0.991 | 0.718 | 0.833 | 0.698 | 0.308 | 0.426 |
| Qwen 2 7B | 0.964 | 0.757 | 0.848 | 0.998 | 0.994 | 0.996 |
| Qwen 2.5 3B | 0.993 | 0.782 | 0.875 | 0.598 | 1.000 | 0.749 |
| Qwen 2.5 7B | 0.982 | 0.967 | 0.974 | 0.907 | 0.972 | 0.939 |
| Qwen 2.5 72B | 0.976 | 0.986 | 0.981 | 0.398 | 1.000 | 0.570 |
| GPT4o-mini | 0.964 | 0.974 | 0.969 | 0.360 | 1.000 | 0.529 |

Table 1: **VLMs can perform anomaly detection**. Results are evaluated using precision, recall, and F1-score across two anomaly detection tasks where anomaly inputs are considered positive cases. In both tasks, VLMs achieve high recall, successfully identifying most anomalous inputs. However, for the ECG task, models exhibit lower overall performance (lower F1), primarily due to low precision caused by over-rejection by frequently abstaining even on valid inputs.

to perform classification on target samples, i.e. those classifiable into CIFAR-10 categories, and to identify and reject the anomalous samples outside CIFAR-10 concepts. This setup is known to be challenging for OOD/anomaly detection methods (Yang et al., 2023b) as the anomalous samples are visually very similar to classifiable samples, since they come from the same dataset. We construct the evaluation dataset with 3,000 images in total, composed of 1,800 classifiable images and 1,200 anomaly ones, where a random classifier would give a precision of 0.4 and a recall of 0.5.

**ECG v.s. Not ECG**: We used ECG signals from the PTB database (Wagner et al., 2020), treat the signals as images, prompting the VLM to classify signals as normal or abnormal, as well as identifying and rejecting anomaly inputs, which we construct with randomly generated line patterns (example shown in the top left panel of Figure 2, detailed in Appendix. C). Our evaluation dataset consists of 500 images each from the normal, abnormal, and anomaly categories, therefore, a random guess baseline should have a precision and recall value of 1/3

Fig. 3 illustrates a representative example from the ECG vs. not ECG experiment. Without explicit instructions to reject anomaly inputs, certain VLMs (e.g., Qwen 2.5 3B) hallucinate answers when presented with random lines, attempting to force classification despite the input clearly not being an ECG. However, when the prompt is augmented with a single line allowing rejection of non-ECG inputs (highlighted in red in the prompt), the same model correctly identifies the input as anomalous and responds with `Unknown`.

The quantitative results are shown in Table. 1, where we evaluate the performance using precision, recall, and F1 value as discussed in Sec. 3.1. For the CIFAR-10 vs. Not CIFAR-10 task, all models achieve high precision larger than 0.96, indicating that models are not showing "over refusal", but with varying recall rates, indicating some models (e.g. Qwen2.5 v.s. Qwen 2) can more reliably identify anomalous inputs when instructed to do so than others. For ECG vs. Not ECG task, overall performance drops significantly as indicated by the lower F1 value. Llama 3.2 11B shows limited ability to identify anomalous ECG inputs (0.308 recall), while Qwen 2 and Qwen 2.5 models demonstrate remarkable precision and recall. Interestingly, GPT4o-mini shows perfect recall but lower precision (0.360), suggesting a tendency toward over-rejecting valid ECG signals as anomalous.

### 4.2 Classification with rejection under ambiguous inputs

For classification with rejection, we consider the following two problems: Classifying corrupted ImageNet images into CIFAR-10 categories and morphological classification of Galaxy Zoo images. The reason behind choosing these two datasets is that they have ground truth ambiguity levels provided, such that we could better understand how VLM behaves as the level of uncertainty varies.

- **Classify ImageNet-C into CIFAR-10 categories** Similar to the anomaly detection setting, we again selected some classes from ImageNet that overlaps with CIFAR-10 categories (detailed in Appendix. B), then we used their corrupted version to introduce ambiguity in the input, we considered 4 types of corruption: Gaussian noise, defocus blur, pixelated, all from ImageNet-C (Hendrycks & Dietterich, 2019),

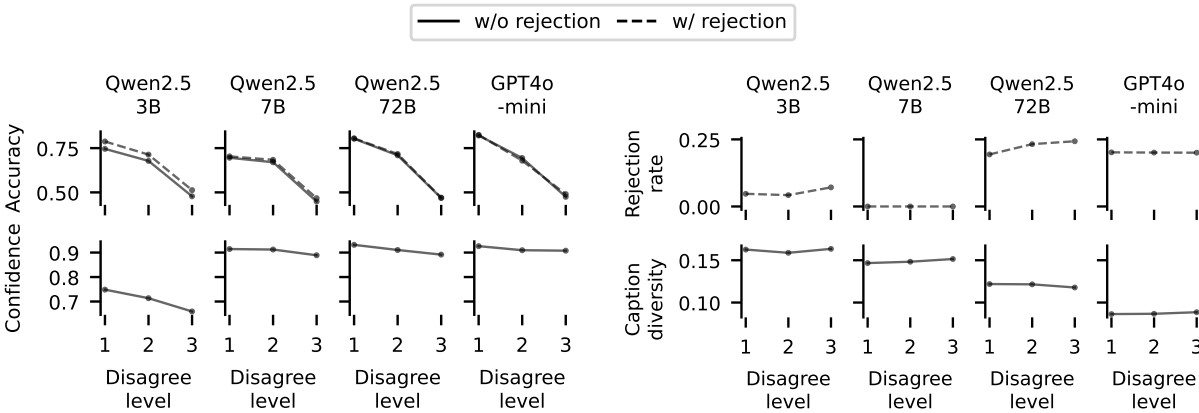

Figure 6: **Classification with rejection fails on Galaxy Zoo.** As annotator disagreement level increases (higher input ambiguity), VLMs' accuracy degrades, but do not effectively utilize rejection to improve performance (top left). Caption diversity and confidence remain *flat* (bottom row), indicating that models fail to recognize the uncertainty in this domain, likely due to insufficient domain knowledge.

- and pixmix from Hendrycks et al. (2022). For all corruption types and levels, we use the same 1,000 randomly selected images.

- **Classification of Galaxy Zoo images** We considered Galaxy Zoo 2, a crowdsourced dataset for galaxy morphological classification. Importantly, each image comes with an annotator agreement score (`leaf_prob` in the metadata table), where a low agreement score implies that an image is ambiguous among classes due to, e.g. unclear visual features. We select a 5,000 sample subset from the dataset and prompt the model to classify them into the four categories provided from Galaxy MNIST (Walmsley, 2022). We present more details on dataset construction in Appendix. D.

**ImageNet-C results** We begin by looking at the VLM's behavior on images corrupted by Gaussian noise, as shown in Fig. 2. Before explicit prompting VLMs to reject ambiguous samples, we first look at VLM's behavior when prompted naively just to classify the image, where we find that some models can already reject highly noised inputs (Fig. 2a) instead of hallucinating an answer, however if we look at their accuracy, we can see that it still drops significantly as corruption level increases (solid line, Fig. 2b). Now when they are *explicitly* prompted to reject ambiguous samples, the accuracy among classified samples becomes much higher, indicating that the models are internally aware of the uncertainty in the input, but it needs to be "activated" via prompting. We observe similar results for other corruption types shown in Appendix. F. Additionally, we studied caption diversity (Sec. 3.2), for each image, we prompt the model to independently generate 20 captions. Broadly, we find that as the corruption level increases, the overall caption diversity increases (Fig. 5 top); additionally, rejected samples are those that receive higher caption diversity (Fig. 5 bottom, red v.s. blue lines), indicating that indeed caption diversity reflects VLMs' internal uncertainty level on input images, aligning with our hypothesis.

**GalaxyZoo results** However, this is not the case for Galaxy Zoo, the results are shown in Fig. 6, where we see that the accuracy significantly drops as ambiguity level increases, but rejection option only provides marginal improvement, indicating that the VLMs do not understand the ambiguity, such result is also predicted if we look at the caption diversity, which stays constant as annotator diagreement level increases, in contrast to the pattern for ImageNet-C images, an illustrative comparison is provided in Fig. 4. Notice that we also study just the output confidence, i.e. we prompt the model to randomly generate multiple answers and look at the maximum softmax probability of the averaged prediction vector, and again, we did not see any sign that the model is aware of the ambiguity. We hypothesize that such task would require more domain specific knowledge with customized image embedding model (Walmsley et al., 2022; Parker et al., 2024), indeed as verified by Wang et al. (2024b), a general purpose CLIP model significantly *underperforms*

fine-tuned CLIP on galaxy classification task, however building a VLM with domain knowledge goes beyond the scope of this work.

**Hypothesis and potential solution for GalaxyZoo's failure**  We hypothesize that the failure of rejection prompting on GalaxyZoo arises from the existence of high epistemic uncertainty. In particular, for natural images, there is low epistemic uncertainty; the model can be prompted to detect samples with high aleatoric uncertainty (corrupted images) and reject them. For Galaxy Zoo images, there is high epistemic uncertainty (due to lack of domain knowledge), and the model cannot distinguish between samples of high / low aleatoric uncertainty (the rejection rate stays constantly low as human annotator disagreement level increases). Intuitively, this suggests that the model needs domain knowledge (low epistemic uncertainty) to understand the ambiguity in the inputs (level of aleatoric uncertainty). This also indicates that if we manage to reduce epistemic uncertainty, e.g., via prompting or few-shot examples, the model may become capable of rejecting samples of high aleatoric uncertainty. Our additional experiment results (Appendix G) verify the feasibility: Instead of directly asking the VLM to perform galaxy classification, we can prompt the model to answer questions in Galaxy Zoo's decision tree (https://data.galaxyzoo.org/gz_trees/gz_trees.html), which decomposes galaxy classification into a sequence of simple visual questions with no domain knowledge of astrophysics needed. This can be viewed as augmenting the model verbally with the knowledge of how to recognize galaxy images. As a proof of concept, we considered the first question in the decision tree (Smooth v.s. Featured galaxy), which decides whether the image falls into (`smooth_round, smooth_cigar`) v.s. (`edge_on_disk, unbarred_spiral`) categories. Broadly, given carefully designed prompts, where we explain the concepts of the categories and rejection conditions in plain language. On instances where human annotators show high disagreement, the model rejects the instance by returning "Unknown" (Figure 10). On instances where human annotators reach a consensus, the model's answer aligns with the annotators' consensus (Figure 11). Note that we do find that the model's rejection behavior is more sensitive to the prompting design in this case compared with the ImageNet-C cases. We believe the prompt sensitivity can be reduced through fine-tuning to internalize the knowledge and conditions for rejection.

## 5  Conclusion

To summarize, our work revisits two classic uncertainty quantification evaluation settings: handling corrupted inputs and anomaly detection, which are challenging for small-scale models trained from scratch due to the inputs' OOD nature. While the boundary of OOD may be less clear for VLMs, we argue that these tasks still represent real-world challenges involving *inherent* data ambiguity and anomalousness, issues that cannot be resolved through model scaling alone. We evaluate VLMs on these two problems and find that, for standard benchmarks (e.g. ImageNet-C and CIFAR-10 vs. Not CIFAR-10), models generally perform well through providing explicit rejection option in prompt.

However, in problems requiring specialized expert knowledge, such as galaxy classification, VLMs consistently exhibit suboptimal performance, which highlights the importance of domain-specific foundation models: These models are not only essential for achieving strong task performance but are also critical for ensuring reliability. Without a proper understanding of the input domain, models struggle to recognize and quantify uncertainty, since a good understanding of the input could be a necessary prerequisite for understanding the uncertainty associated with it.

## 6  Limitation and future work

**Evaluation on reasoning VLMs**  The current evaluation is limited to non-reasoning VLMs. Future work may consider how long chain of thoughts may alter the VLM's robustness, mimicking studies in the pure language domain (Kirichenko et al., 2025).

**Extended experiments on Galaxy Zoo**  The current evaluation (Appendix G) on integrating domain knowledge via Galaxy Zoo's decision tree is limited to the proof-of-concept stage. Future work can consider further extending this setup, as well as exploring the idea of decomposing difficult tasks into a tree of simpler tasks in a broader range of AI4Sci problems.

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

# A Prompt

## A.1 Anomaly detection

### CIFAR-10 v.s. not CIFAR-10

```
You are given an image, check if the image is from one of the following 10 classes
:
airplane, automobile, bird, cat, deer, dog, frog, horse, ship, and truck, if so,
classify the image into one of 10 classes, if not, label them as "unknown".

Think step by step, and give an answer in the following format:
Reasoning: <your_reasoning>
Answer: <your answer>
```

### ECG v.s. not ECG

```
Your task is to classify the given ECG signal into the normal case and the
abnormal cases.

Note that if the signal does NOT look like ECG signal or cannot be classified into
either of the categories, return ''unknown'' as the answer.

Give an answer and step-by-step reasoning in the format:
Reasoning: <your_reasoning>
Answer: normal/abnormal/unknown
```

## A.2 Classification with rejection under ambiguous inputs

### ImageNet-C

```
You are given an image, the image belongs to one of the following 10 classes:
airplane, automobile, bird, cat, deer, dog, frog, horse, ship, and truck.

Please assign each an image a label from the 10 classes, think step by step, and
give an answer in the following format:
Reasoning: your_reasoning
Label: class_name

Notice that if you find an image very ambiguous and cannot confidently classify it
, return "unknown" as the label.
```

### Galaxy Zoo classification

```
You are an expert astronomer specializing in galaxy morphology. You will be shown
images of galaxies and need to classify them into one of the following categories
by analyzing their visual characteristics:

1. smooth_round: Galaxies that appear completely or nearly circular, with a smooth
 decrease in brightness from center to edge.
2. smooth_cigar: Galaxies that appear elongated and smooth, with an elliptical
shape resembling a cigar, showing a gradual decrease in brightness from center to
edge.
3. edge_on_disk: Galaxies viewed from the side, appearing as a thin line or disk
with a bright central bulge, similar to viewing a dinner plate from its edge.
4. unbarred_spiral: Spiral galaxies without a central bar structure, showing
distinct spiral arms emanating directly from the galactic center.
```

```
Classify the galaxy into exactly one of the four categories listed above with step
 by step reasoning.
Notice that if you find an image very ambiguous and cannot confidently classify it
,
return "unknown" as the label.

Your response should be structured as:
Reasoning: [Brief explanation of the key visual features that support this
classification]
Answer: [category_name] or unknown
```

### A.3  Caption generation

**ImageNet-C**

```
You are given an image, please generate a short description of the image.
```

**Galaxy Zoo**

```
You are an expert astronomer specializing in galaxy morphology. Please generate a
concise description of the image that describes its key visual characteristics.
```

# B  Mapping between CIFAR-10 and ImageNet classes

In the main text, we extensively consider the setting where we pick images from ImageNet (or its corrupted version) that overlap with CIFAR-10 categories and prompt VLMs to classify the images into CIFAR-10 categories. The exact mapping is presented in Table. 2. Notice that here we ignored the deer and horse categories from CIFAR-10 as we cannot find exact mapping in ImageNet categories.

Table 2: Mapping from CIFAR-10 categories to corresponding ImageNet classes (ImageNet class indices in parentheses).

| CIFAR-10 Category | ImageNet Classes (Indices) |
|---|---|
| Airplane | Airliner (404), Warplane (895) |
| Automobile | Beach Wagon (436), Convertible (511), Model T (661), Sports Car (817) |
| Bird | Jay (10), Magpie (11), Eagle (12), Vulture (13), |
| | Additional Birds (92, 93, 94, 95, 96) |
| Cat | Tabby Cat (281), Tiger Cat (283), Persian Cat (284), Siamese Cat (285) |
| Dog | Chihuahua (151), [Every 20th dog class up to] Mexican Hairless (268) |
| Frog | Bullfrog (30), Tree Frog (31) |
| Ship | Container Ship (510), Liner (628), Pirate Ship (724), |
| | Schooner (780), Submarine (833) |
| Truck | Fire Truck (555), Garbage Truck (569), Moving Van (675), Pickup Truck (717), |
| | Police Van (734), Tow Truck (864), Trailer Truck (867) |

## C  Anomaly ECG inputs construction

To construct anomaly inputs that are clearly not ECG signal, we use a two step routine: We begin by randomly choose a random number generator

```
def get_random_number(size=128):
    rng_list = [
        np.random.normal, np.random.gamma, np.random.exponential,
        np.random.poisson, np.random.uniform, np.random.chisquare,
        np.random.geometric
    ]
    rng = np.random.choice(rng_list)
    return rng(0.6, size=size)
```

Then we generate a random line figure, with 128 points, as the anomaly inputs via

```
plt.plot(get_random_number(128), get_random_number(128), linewidth=0.8, c='black')
```

## D  Galaxy Zoo explanation

Galaxy Zoo is a citizen science dataset, for each galaxy image, annotators are asked to follow the decision tree (https://data.galaxyzoo.org/gz_trees/gz_trees.html) and perform classification for the galaxy image based on the visual features. Importantly, the metadata of the dataset provides a property called `leaf_prob` that describes the agreement level among annotators for the mostly agreed category. We later use `leaf_prob` as a score that denotes the inherent ambiguity level of a given galaxy image.

To actually construct the dataset for evaluation VLM, we first select all images whose majority vote label falls into one of the following categories:

`smooth_round, smooth_cigar, edge_on_disk, unbarred_spiral`

which are the four representative galaxy categories studied in Galaxy MNIST (Walmsley, 2022). Then we *randomly* selected 5,000 images from the pool.

Then, based on the `leaf_prob`, we create three levels of disagree level

```
Leaf prob in (0.75, 1.0): Disagree level 1
Leaf prob in (0.5, 0.75): Disagree level 2
Leaf prob in (0.0, 0.5) : Disagree level 3
```

The distribution of categories and `leaf_prob` from the 5,000 samples are plotted in Fig. 4b

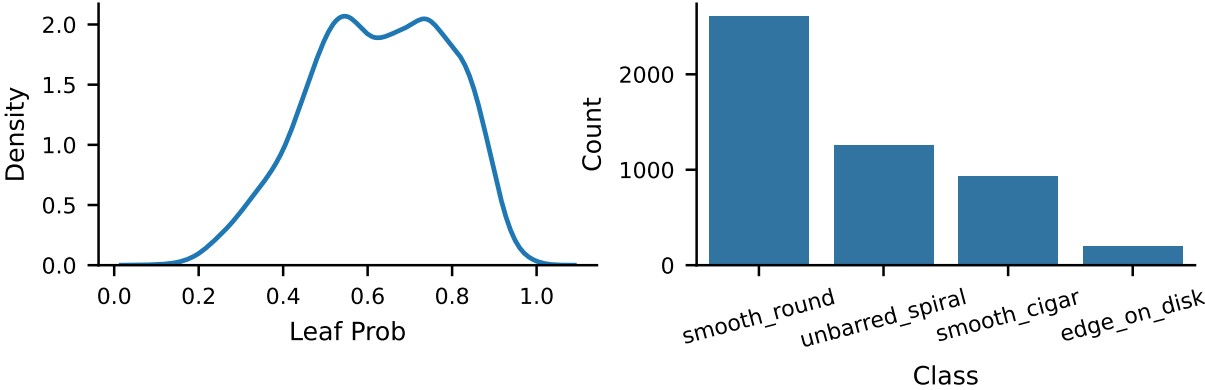

Figure 7: Statistics of the 5,000 sample galaxy zoo datasets used in the experiments.

# E Ablation study: Effect of instruction prompt

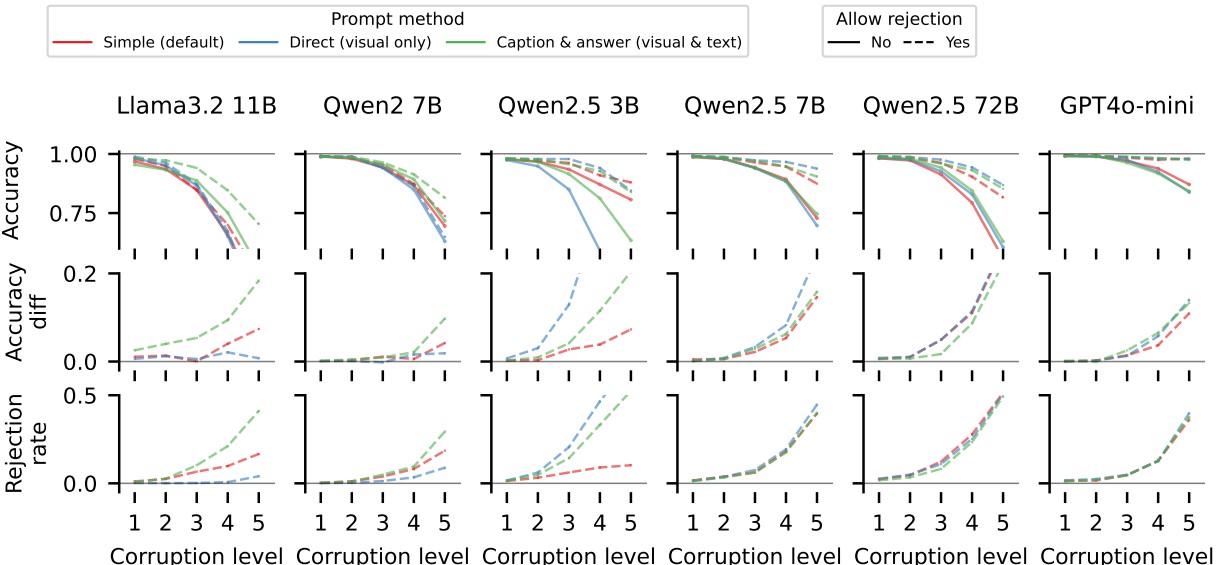

Figure 8: **Ablation study on the prompting strategy.** Similar to the setting studied in Fig. 2b, here we consider different prompting strategies (line color, detailed in Appendix. E), and models (columns). The second row illustrates the accuracy improvement, we can see that for certain old models, the prompting strategy matters, but for other, especially the newer ones, it matters little.

| Model | Simple | | Direct | | Caption & Answer | |
|---|---|---|---|---|---|---|
| | **Precision ↑** | **Recall ↑** | **Precision ↑** | **Recall ↑** | **Precision ↑** | **Recall ↑** |
| Llama 3.2 | 0.991 | 0.718 | 0.994 | 0.492 | 0.994 | 0.897 |
| Qwen 2 | 0.964 | 0.757 | 0.994 | 0.215 | 0.992 | 0.917 |
| Qwen 2.5 3B | 0.993 | 0.782 | 0.994 | 0.549 | 0.993 | 0.725 |
| Qwen 2.5 7B | 0.982 | 0.967 | 0.992 | 0.837 | 0.992 | 0.972 |
| Qwen 2.5 72B | 0.976 | 0.986 | 0.969 | 0.992 | 0.987 | 0.987 |
| GPT4o-mini | 0.964 | 0.974 | 0.974 | 0.975 | 0.988 | 0.974 |

Table 3: Similar to the setting in Table. 1, but here we considered different prompting strategy, as discussed in Appendix. E. Overall, smaller and older models such as Llama 3.2 and Qwen 2 are sensitive to the prompting strategy, more powerful models are less sensitive (e.g. GPT4o-mini and Qwen 2.5 72B).

We perform an ablation study where we vary the prompt that instructs the VLM to answer the query in a certain way. To be more specific, we considered the following three regimes:

- **Simple** Simple and standard way: we prompt the model to provide step-by-step reasoning (Wei et al., 2022) and then provide a classification answer.
- **Direct** We prompt the model to *only output the classification answer*, which prohibits the model from generating any explicit intermediate verbal reasoning steps, including image caption.
- **Caption & answer** We explicitly prompt the model to *always first caption the image*, then answer the question using both the caption and the image.

The goal is to understand whether:

- The model can perform the uncertainty quantification task through only reading the image (Direct prompting)
- The model cannot understand the uncertainty by just reading the image, but would need to first caption the image and then utilize the textual description to determine the level of ambiguity.

In Fig. 8, we present accuracy with and without the rejection option (solid and dashed lines) in the first row v.s. corruption level, the improvement of allowing the rejection option (second row) and the rejection rate (third row) under different models (columns) and prompts (line colors). Particularly, for Llama 3.2 and Qwen2, the direct prompt (blue lines) shows almost no improvement when rejection option is enabled and a low rejection rate, and the models only start to reject ambiguous inputs under **Caption & answer** (green line), whereas for other models, the VLMs demonstrate the ability of rejection under all types of prompts. This implies that these two models (Llama 3.2 and Qwen2) lack the ability to "silently" reason about uncertainty, instead, they need to to first (be asked to) spell the image out, then use the textual feature to determine uncertainty. The observation is again confirmed in the OOD detection task (Table. 3), the metrics under the direct prompt for these two models are outperformed by the Caption & answer prompt.

Regardless, the ability of rejection from the latest models does not seem to be affected by the prompting strategy. It is also worth noting that, the fact that these models can "silently" quantify uncertainty does not rule out the possibility that they are verbal thinker internally: They could be "implicitly" using verbal reasoning inside.

In the blocks below, we provide example outputs from Llama 3.2 and Qwen2.5 7B given a Gaussian noise corrupted ship image, where Llama 3.2's output varies significantly with prompts whereas Qwen2.5 7B shows much higher stability

**Llama3.2**

```
Direct:
Label: cat<|eot_id|>
-------------------------------
Simple:
**Problem Statement:**
We are given an image and asked to classify it into one of the following 10
classes: airplane, automobile, bird, cat, deer, dog, frog, horse, ship, and truck.

**Approach:**
To solve this problem, we will analyze the image step by step and make an educated
 guess based on the visual features present in the image.

**Step 1: Observe the Image**
The image appears to be a low-resolution, pixelated photograph of a vehicle or an
animal. The image is too blurry to make out any distinct features.

**Step 2: Look for Clues**
Upon closer inspection, we notice that the image has a dark body with a long,
curved shape protruding from it. This could be a tail or a limb.

**Step 3: Eliminate Impossible Classes**
We can eliminate the classes that do not match the visual features observed in the
 image. For example, we can eliminate "airplane" and "ship" because they do not
have a curved shape or a dark body.

**Step 4: Make an Educated Guess**
Based on the remaining classes, we can make an educated guess. The image resembles
 a cat, which has a dark body and a long, curved tail.

**Answer:**
Reasoning: The image is too blurry to make out any distinct features, but it
resembles a cat.
Label: cat<|eot_id|>
-------------------------------
Caption & answer:
The image appears to be a pixelated representation of a building, possibly a
skyscraper or high-rise structure. The dominant colors are shades of gray, with
```

```
some red and green hues visible in the upper right corner. The overall effect is
one of a highly distorted or low-resolution image, making it difficult to discern
any specific details.

**Classification:**
Based on the description, I would assign the image the label "unknown" as it does
not clearly resemble any of the specified classes (airplane, automobile, bird, cat
, deer, dog, frog, horse, ship, or truck).<|eot_id|>
```

**Qwen 2.5 7B**

```
Direct:
unknown
-------------------------------
Simple:
Reasoning: The image appears to be a close-up of a textured surface with various
colors and patterns. It does not resemble any of the typical shapes or features
associated with the 10 classes provided (airplane, automobile, bird, cat, deer,
dog, frog, horse, ship, and truck). The image is too abstract and lacks distinct
characteristics that would allow for confident classification into one of these
categories.

Label: unknown
-------------------------------
Caption & Answer:
Description: The image appears to show a close-up view of a colorful, patterned
surface with geometric shapes and vibrant colors. The shapes seem to be arranged
in a grid-like pattern, possibly resembling tiles or a mosaic.

Label: unknown
```

# F Full results across all corruptions

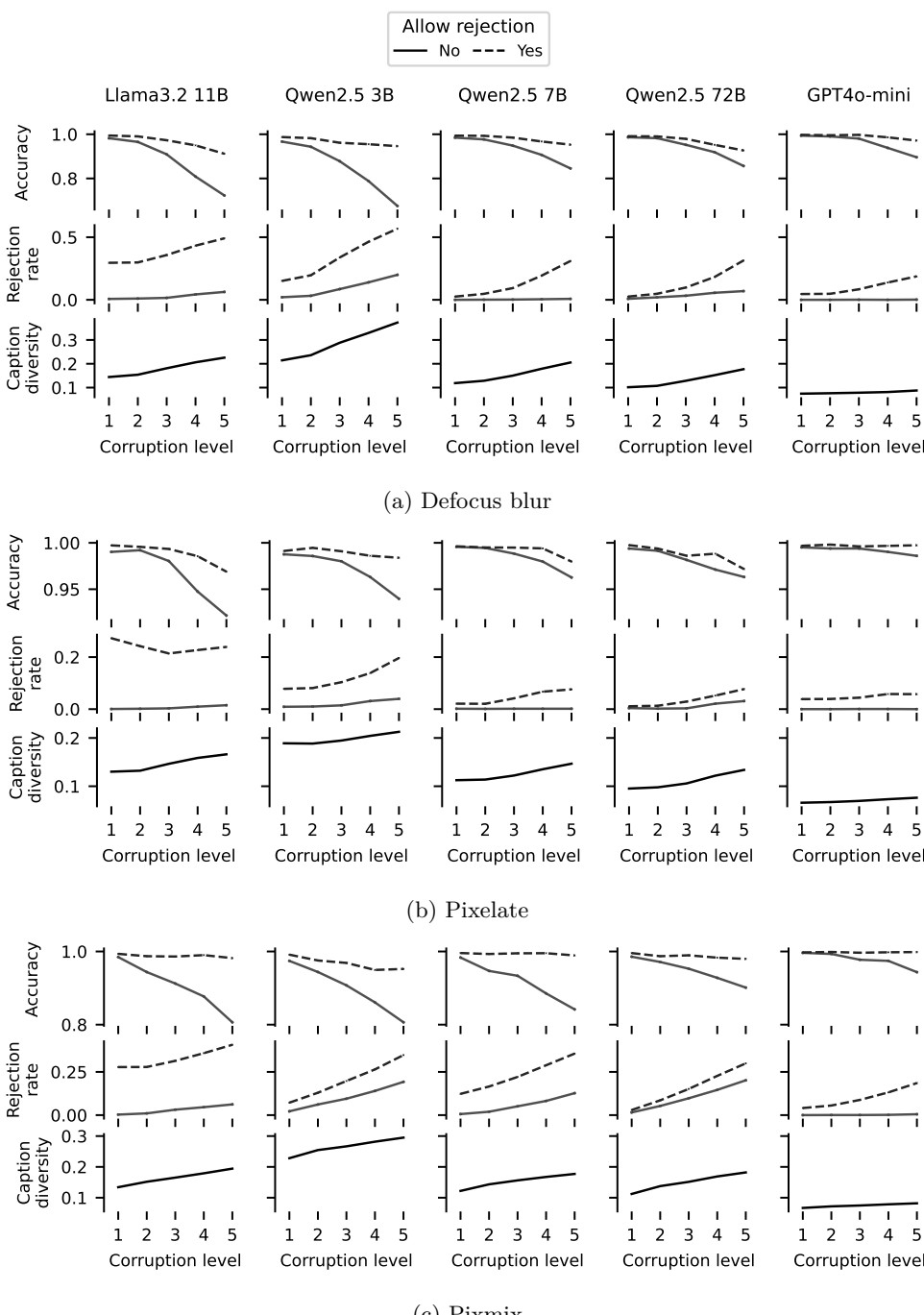

Figure 9: Similar to the setting in Fig. 2b, but under different corruption types. The observation still holds: Enabling rejection allows VLM to abstain ambiguous samples, providing nearly perfect accuracy for classified samples. Additionally, caption diversity, as we predicted in Sec. 3.2, steadily increases as corruption level intensifies.

## G   Extended result on GalaxyZoo

We prompt the VLM (Qwen2.5 7B) to answer the first question in the decision tree using the following prompt, where the red part denotes the prompt for rejection

```
Select one of these categories that best describes the image:

- features or disk: The image has any features at all, such as spiral arms, bars, dust
lanes.
- smooth: The image gradually fades in all directions from the center (it can have
tight bright core).

**Notice that** if you see features but cannot interpret the structure with confident,
return "Unknown".
**Notice that** if you are unsure between the options, return "Unknown".

Think step by step, answer the question in the following format:
Explanation: <your explanation here>
Answer: <your answer here>
```

In Figure. 10 and Figure. 11, we present example images with low and high consensus between annotators, respectively, for the first question in the decision tree (https://data.galaxyzoo.org/gz_trees/gz_trees.html). Broadly, we find that, unlike the observations in the main text, where the model fails to reject high disagreement images, now the VLM becomes capable of rejecting inputs for which no consensus is reached thanks to the simplicifcation of tasks and domain knowledge provided in the prompt.

P(Smooth): 0.40
P(Featured): 0.60

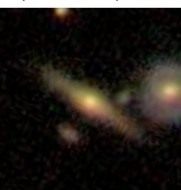

Explanation: The image appears to show a galaxy with a central bright region and some fainter structures extending outward. There are hints of spiral arms or other features, but they are not clearly defined or easily interpretable.
Answer: Unknown

P(Smooth): 0.59
P(Featured): 0.41

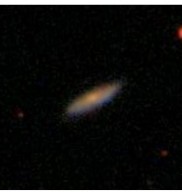

Explanation: The image shows a galaxy with a distinct elongated shape and a bright central region, which is characteristic of an elliptical galaxy. There are no visible spiral arms, bars, or dust lanes that would suggest it is a spiral galaxy. The brightness gradient is not smooth; instead, there is a concentration of light towards the center, which is typical for elliptical galaxies.
Answer: Unknown

P(Smooth): 0.67
P(Featured): 0.33

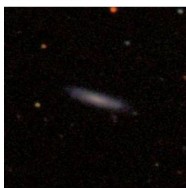

Explanation: The image shows a galaxy with a distinct elongated shape and a faint glow extending along its length. There are no clear spiral arms, bars, or dust lanes visible in this image. The brightness is relatively uniform across the galaxy, without any noticeable bright core or fading pattern.
Answer: Unknown

Figure 10: On images where annotators show disagreement (annotator voting probability shown on the top of each image), VLM also rejects these samples by answering "Unknown".

P(Smooth): 0.95
P(Featured): 0.05

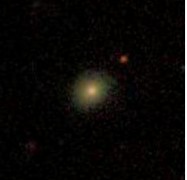

Explanation: The image appears to show a galaxy with a smooth, elongated shape and a central bright region. There are no visible spiral arms, bars, or dust lanes that would suggest the presence of features typical of galaxies with active star formation or recent interactions.
Answer: smooth

P(Smooth): 0.89
P(Featured): 0.11

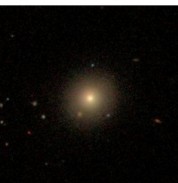

Explanation: The image appears to show a galaxy with a central bright region and a fainter halo extending outward. There are no distinct spiral arms, bars, or dust lanes visible in the image. The brightness is more uniform across the galaxy, suggesting it might be a smooth distribution rather than having specific structural features.
Answer: Smooth

P(Smooth): 0.92
P(Featured): 0.08

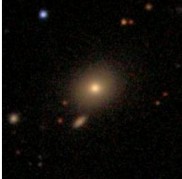

Explanation: The image appears to show a galaxy with a central bright region and a fainter halo extending outward. There are no distinct spiral arms, bars, or dust lanes visible in the image. The brightness is concentrated in the center, which suggests a smooth distribution of light.
Answer: Smooth

P(Smooth): 0.19
P(Featured): 0.81

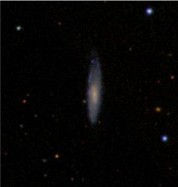

Explanation: The image shows a galaxy with a distinct elongated shape and a central bright region, which is characteristic of a spiral galaxy. The structure appears to have spiral arms extending outward from the central bulge.
Answer: features or disk

P(Smooth): 0.12
P(Featured): 0.88

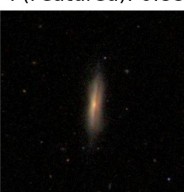

Explanation: The image shows a galaxy with a distinct spiral structure. There are visible spiral arms extending outward from the central region, which is characteristic of spiral galaxies. The presence of these spiral arms indicates that the galaxy has features.
Answer: features or disk

P(Smooth): 0.03
P(Featured): 0.97

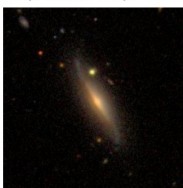

Explanation: The image shows a galaxy with a distinct spiral structure, which includes spiral arms extending outward from the central region. There is also a bar-like feature running through the center of the galaxy, which is a common structural element in many galaxies.
Answer: features or disk

Figure 11: On images where annotators show consensus (annotator voting probability shown on the top of each image), VLM returns answers aligned with the majority vote results.

## H   Licenses for existing assets

**Datasets:**

- **ImageNet / ImageNet-C**
  License: Custom ImageNet Terms of Use (Non-commercial research only)
  URL: https://www.image-net.org/download
- **CIFAR-10**
  License: MIT License
  URL: https://www.cs.toronto.edu/~kriz/cifar.html
- **Galaxy Zoo / Galaxy MNIST**
  License: Creative Commons Attribution-NonCommercial-ShareAlike (CC BY-NC-SA 4.0)
  URL: https://data.galaxyzoo.org/
- **PTB-XL (ECG Dataset)**
  License: PhysioNet Credentialed Health Data License (Restricted; requires credentialed access)
  URL: https://physionet.org/content/ptb-xl/

**Models:**

- **GPT-4o-mini**
  License: Proprietary (OpenAI Terms of Use)
  URL: https://openai.com
- **Llama 3.2 (Meta)**
  License: Meta Llama 3 Community License (Non-commercial research use only)
  URL: https://ai.facebook.com/resources/models-and-libraries/llama
- **Qwen 2 / Qwen 2.5 (Alibaba)**
  License: Qwen License Agreement (Permits research and certain commercial uses)
  URL: https://qwen.aliyun.com/

