# OpenReview forum: "Are vision language models robust to classic uncertainty challenges?"
_TMLR — Accepted by TMLR_

### Review · Reviewer_egu1 · 2025-12-15

**Summary Of Contributions:**

**Summary**

This paper claims that today's VLMs, despite scaling in both data and parameters compared to prior approaches, still suffer from weakness to both covariate and concept shift, and cannot explicitly tag such shifts as uncertain, rather than providing a misclassification. Explicitly:

> While VLMs may no longer face clear-cut in-distribution v.s. out-of-distribution boundaries, they still encounter practical challenges in handling visually ambiguous inputs, or anomaly inputs that fall outside the semantic scope defined by a user prompt.

The paper studies how VLMs behave through several case studies: (1) corrupted ImageNet images, (2) CIFAR-10 vs. Non CIFAR-10 images, and (3) two specialized tasks, ECG signal classification and Galaxy image recognition. They come to the conclusion that modern models are robust under many situations, but still struggle in others.

The paper shows the following findings:
* In anomaly detection, the paper asks VLMs to predict semantic mismatches, first by taking CIFAR-10 vs. non CIFAR-10 classes, which the models achieve with quite high F1 scores, and then ECG vs. not ECG signals (as images), on which the models are asked to predict which signals are, and are not, valid ECG. In general, the paper shows that VLMs are quite capable of anomaly detection when prompted to do so.
* The paper also asks VLMs to perform classification under ambiguous inputs (ImageNet-C into CIFAR-10, GalaxyZoo). The paper shows  in F.2 that as corruption level increases, accuracy generally falls, and rejection rate rises. In all cases, it appears that explicit prompting for uncertainty rejection appears to help. In F.6, using annotator disagreement as a proxy for ambiguity, we see that VLM accuracy degreades, and VLMs do not naturally use rejection to improve performance rates (likely due to insufficient domain knowledge).
* The paper finds that generated caption diversity is correlated with corruption, and input ambiguity.

**Audience:**

Yes

**Audience Explanation:**

This experiment, while limited, does not have any clear structural concerns. The experiments themselves are novel and structured. The clearest finding is:

> prompting the models explicitly to "reject ambiguous inputs" substantially enhances their reliability

which is an unambiguous, and well-supported finding that some people may find interesting. If the introduction were scoped correctly, the findings are at least interesting as a building block to larger uncertainty.

**Broader Impact Concerns:**

I have no ethical concerns with the implications of this work - and in my opinion a broader impact section is not required.

**Claims And Evidence:**

No

**Claims Explanation:**

The primary technical contribution is based on the  the observation that VLMs uncertainty level about an input image can be revealed by prompting the VLM to generate multiple captions under random sample decoding, where VLMs tend to generate more diverse captions for visually ambiguous samples. This claim, while factually supported, in my opinion is somewhat misguided. VLMs will produce more diverse captions under at least two scenarios: the first, which the paper correctly identifies, is when the image is OOD (grounded in Figure 4a and 5.  The second, which the paper ignores, is when the captions **require** uncertainty. In many cases, there are multiple ways to correctly describe an image, which will lead to higher caption diversity naturally under sampling. In these cases, we are penalizing a model for flexibility that is actually desired, rather than uncertainty which should be expressed, see [1].

Some other claims are also overly broad:

* "Neural networks often struggle with input uncertainty due to their training methodology. When trained on carefully curated datasets with minimal ambiguity, models develop a tendency to produce high-confidence predictions for all inputs, regardless of their clarity or relevance." (Sec. 2.1) I don't know of any work which has clearly attributed input uncertainty to training methodology - in general, I would instead expect this to be a consequence instead of the data, where there isn't sufficient examples of uncertainty in the data to avoid biasing the model to produce a confident answer. This, to me, seems like an unnecessary claim that could be removed.
* The reasons at the end of page 3/the arguments on page three are actually quite poor arguments for a lack of uncertainty classification. In fact, since many times uncertainty is data-centric, to me at least, there's no expectation that model scaling will improve uncertainty estimation.

Finally, there are several statements in the introduction that appear to be motivation, but are not resolved in the paper:

* "We believe our study demonstrates an important lesson: Despite the clear superiority of modern, large-model AI systems, do not assume they solve older benchmarks." From what I can tell, the performance on older benchmarks is competitive with, if not exceeds performance of older methods? What is the intent here?
* "Are the traditional evaluation tasks of corruption robustness and OOD detection meaningful for VLMs?" I don't feel like this paper makes a stance on this claim, or provides sufficient evidence to determine either way if this is the case.
* The paper makes claims about all VLMs, but tests a very minimal range of older VLMs (Llama 3.2, Qwen 2, GPT-4o-mini), and no reasoning models, which are (in my personal observation), much better at understanding and reasoning over uncertainty.

[1] Chan, David, et al. "Ic3: Image captioning by committee consensus." Proceedings of the 2023 Conference on Empirical Methods in Natural Language Processing. 2023.

**Requested Changes:**

Some clear changes are, in my opinion, necessary:
* Narrow/Reframe the caption diversity claim: Caption diversity should probably be explicitly reframed as a diagnostic proxy, rather than a general uncertainty estimator (and the difference between sources of diversity should be specified). The claim should also probably be restricted to classification-oriented prompts
* Remove/weaken the overly broad training-methodology claims: one of the things that needs to be discussed is the connection between uncertainty and data.
* Clarify the Paper’s Position on “Are Classic Benchmarks Solved?”: I think that this paragraph needs a second look in terms of the overall claims, and what is meant by "solved" - perhaps we mean "robust by default?"
* The paper makes claims about “VLMs” broadly but evaluates a narrow subset, excluding reasoning-focused models, the set should be expanded, or the claims rephrased.
* Human baselines/baselines in the tables themselves would be appreciated, to help ground overall performance.

Additionally, the paper would benefit from clearer alignment with recent work framing uncertainty as primarily data- and task-dependent rather than a consequence of insufficient model scale, as well as a more explicit interpretation of the Galaxy Zoo failure as evidence that uncertainty awareness depends on semantic grounding. Reordering the contributions to foreground prompted abstention as the main result, with caption diversity treated as a secondary diagnostic, would also improve clarity and better reflect the strength of the empirical findings.

---

> ### Author Response · Authors · 2026-01-05
> **Author response (1/2)**
>
> We would like to thank the reviewer for the detailed review. Please see below for our responses.
>
> # Caption diversity
>
> > Caption diversity should probably be explicitly reframed as a diagnostic proxy, rather than a general uncertainty estimator
>
> Yes, we do agree that caption diversity is mainly used as a diagnostic proxy for checking whether the VLM understands the ambiguity in the images, rather than a generic uncertainty estimator, as we pointed out in the title of Sec 3.2, where we describe caption diversity as a tool for understanding the underlying mechanism of rejection.
> However, caption diversity is not tied to the classification task, since the caption is query-independent.
>
> Regarding different sources of uncertainty, first of all, we would like to clarify that the visually ambiguous samples are not OOD samples, but rather, in our opinion, images with unclear visual features that result in multiple ways to correctly describe, e.g. if the car corrupted image in Figure 4 is provided to annotators for captioning, we believe the annotators will provide a diverse set of captions.
>
> However, we do agree that when there is too much content in the images to be described in a small fixed-length caption, e.g., in the examples shown in [1], the VLM may also generate diverse captions, although the input is not necessarily visually ambiguous. We have included an additional clarification at Sec 3.2 (highlighted in red), clarifying that visually ambiguous images are not the only inputs that will trigger uncertain captions.
>
> [1] Chan, David, et al. "Ic3: Image captioning by committee consensus." Proceedings of the 2023 Conference on Empirical Methods in Natural Language Processing. 2023.
>
>
>
> # Arguments related to training methodology and model scale
>
> We would like to clarify that the training methodology is composed of multiple factors, including
> - The training data
> - The training algorithm
>
> We agree that the training data plays a very critical role in uncertainty, as we already pointed out "...When trained on *carefully curated datasets with minimal ambiguity, models develop*...".
> Empirical results have demonstrated that if you train with soft label inputs, acquired either through human annotations [1], or data augmentations [2], the resulting model would show better uncertainty quantification under a standard training regime (i.e., empirical risk minimization).
>
> However, given a dataset without uncertainty, the training algorithm also plays an important role in uncertainty quantification.
> For example, instead of using empirical risk minimization, we can perform posterior inference over the model weights and construct a Bayesian neural network. The resulting model, despite never being exposed to high uncertainty inputs, often shows much better uncertainty quantification.
>
> Our argument around scaling is also related to the training methodology:
> As the model scales larger, the training dataset sizes increase as well, gradually shifting from multi-epoch training on an overparameterized model, where all the training data are memorized, to one-epoch training on an underparameterized model, where the same training sample never appears twice.
> Such an one-epoch training paradigm has been shown to result in better generalization, calibration, and uncertainty quantification ([3]; page 39 in [4]; [5]; [6]).
>
>
> [1] https://arxiv.org/abs/1908.07086
>
> [2] https://arxiv.org/abs/1905.11001
>
> [3] https://arxiv.org/abs/2511.04869
>
> [4] https://preetum.nakkiran.org/pdf/aspen_calibration_full.pdf
>
> [5] https://arxiv.org/abs/2411.14478
>
> [6] https://arxiv.org/abs/2207.05221

---

> > ### Author Response · Authors · 2026-01-05
> > **Author response (2/2)**
> >
> > # Position of the paper
> >
> > > Clarify the Paper’s Position on “Are Classic Benchmarks Solved?”: I think that this paragraph needs a second look in terms of the overall claims, and what is meant by "solved" - perhaps we mean "robust by default?"
> >
> > Yes, by "solved" we mean "robust by default", we have included additional clarification in the last paragraph of Section 1 (highlighted in red).
> >
> >
> > # The paper makes claims about “VLMs” broadly but evaluates a narrow subset, excluding reasoning-focused models, the set should be expanded, or the claims rephrased.
> >
> > Our results show that newer VLMs such as Qwen2.5 and GPT-4o-mini, despite not being reasoning VLMs, are already capable of showing good uncertainty quantification; therefore, we believe existing reasoning VLMs (e.g. Qwen3 / GPT-5) and those that come out in the future, which are most likely to be reasoning models as well, will continue to be robust to the setting we are studying.
> >
> > # Human baselines/baselines in the tables themselves would be appreciated, to help ground overall performance.
> >
> > Thanks for raising this point. We agree that having human performance as a baseline would better contextualize the numbers we reported. Unfortunately, we don't have access to the resources for collecting human annotations at this moment.
> >
> > # Order of contributions
> >
> > > ... Reordering the contributions to foreground prompted abstention as the main result, with caption diversity treated as a secondary diagnostic, would also improve clarity and better reflect the strength of the empirical findings.
> >
> > Thank you for raising this very good point! In the original manuscript, we are already practicing such an order, e.g., in the intro section, we discuss caption diversity as the second-to-last paragraph. We are open to more suggestions from the reviewer to further clarify the contribution of our work.

---

### Review · Reviewer_GM8A · 2025-12-19

**Summary Of Contributions:**

**Summary**

The paper investigates if contemporary large VLMs have effectively solved the classic challenges of anomaly detection (OOD/label space mismatch) and classification under ambiguity (corruption/covariate shift). The authors find that prompting the VLMs to explicitly reject ambiguous inputs leads to better correctness, while a standard QA prompt leads to guessing & hallucinations.

The paper evaluates closed-source model GPT-4o-mini, and open weight VLM families Llama-3.2, Qwen2VL & Qwen2.5VL on two tasks i) Anomaly detection on CIFAR-10 vs not CIFAR-10 & ECG vs  not ECG, where the model is evaluated on ability to reject images that don't belong to any GT category ii) Classification on ambiguous inputs with rejection on ImageNet-C & Galaxy Zoo 2 datasets, where accuracy is computed on samples that are not rejected.  For domains requiring expert knowledge, eg, Galaxy Zoo, reject options doesn't show performance gains, indicating that abstention does not work for domains where the model's internal uncertainty estimates may be poorly calibrated. Finally, they propose to use caption diversity as a proxy for the VLM's uncertainty on an input. Many text descriptions of the i/p image are sampled from the VLM and one minus avg. pairwise cos similarity of all description embeddings gives the caption diversity score. For ImageNet-C images, more corrupt images have higher caption diversity. However this diversity is not observed for the ambiguous samples from the galaxy classification task.


**Strengths**
\+ The ability to handle ambiguous queries is critical for real-world use of VLMs. The paper shows that a simple system prompt change can enable VLMs of various strengths to handle ambiguous inputs. This simple solution makes use of recent advancements in foundation models, where relatively smaller models too have strong instruction-following capabilities and robust world knowledge.
\+ The paper is well written and motivated.
\+ The proposed caption diversity metric makes intuitive sense for images with occlusions, corruptions and other sources of visual ambiguity.


**Weaknesses**
1. How do VLMs handle more nuanced types of ambiguity? For instance question-conditional ambiguity, where the image is clear but the question is underspecified (color of the bottle when the image has many bottles), or when the object of interest is occluded? In such cases perhaps clarification can help the VLM answer the question.
2. For question based semantic ambiguity, it is expected that caption diversity remains low, as a general image description would be relatively unchanged. A discussion on this would better highlight the limitations.
3. For open weight models caption diversity is an expensive metric to compute. A comparison with common logit-based baselines such as answer token entropy (or first token entropy) would strengthen the metric.
4. Although the paper shows rejection prompting to be a good solution, it provides less control on when a sample should be rejected (as observed in the ECG task). It would be interesting to see if different prompting strategies could offer a better tradeoff between accuracy and rejection rates.
5. The failure of rejection prompting on Galaxy Zoo can be seen as due to epistemic uncertainty, where the model lacks sufficient domain knowledge. It is unclear if rejection prompting inherently performs poorly under high epistemic uncertainty. Exploration of simple solutions such as in-context examples in the prompt can help understand this better.

**Audience:**

Yes

**Audience Explanation:**

The paper addresses a practical problem on VLM hallucinations under ambiguous inputs. The paper shows that rejection prompting works well for general images but fails in cases of specialized domains, which is an actionable insight in practice.

**Broader Impact Concerns:**

There are no major concerns related to the work's broader impact.

**Claims And Evidence:**

Yes

**Claims Explanation:**

The claims are well supported by the experimental results. Rejection prompting improves performance on both tasks. The paper highlights the limitation of this approach in domains that require expert knowledge.

**Requested Changes:**

\- [required] Clarify limitations of rejection prompting w.r.t epistemic uncertainty. As mentioned in weakness 5, it remains unclear if rejection prompting only works for aleatoric uncertainty. This clarification is important to understand the limitations of the method.
\- [optional] Expanding discussion to additional, realistic forms of ambiguity seen in VQA would strengthen the paper. A discussion of applicability of caption diversity under such ambiguity types can highlight its applicability.
\- [optional] Compare caption diversity with simpler logit-based uncertainty baselines (as mentioned in weakness point 3).
\- [optional] Can few-shot in-context examples improve performance in Galaxy Zoo (weakness point 5)?

---

> ### Author Response · Authors · 2026-01-05
> **Author response (1/2)**
>
> We would like to thank the reviewer for the detailed review. Please see below for our response to each individual point.
>
> # Rejection prompting w.r.t epistemic uncertainty (Weakness 5, Requested Changes 1 & 4)
>
> Thanks for raising this very good point. We fully agree with the argument that *rejection prompting would fail under high epistemic uncertainty.*
> In fact, this argument fully aligns with our observation
>
> - For natural images, there is low epistemic uncertainty; the model can be prompted to detect samples with high aleatoric uncertainty (corrupted images) and reject them.
> - For Galaxy Zoo images, there is high epistemic uncertainty (due to lack of domain knowledge), and the model cannot distinguish between samples of high / low aleatoric uncertainty (the rejection rate stays constantly low as human annotator disagreement level increases).
>
> Intuitively, this suggests that the model needs domain knowledge (low epistemic uncertainty) to understand the ambiguity in the inputs (level of aleatoric uncertainty).
> This also indicates that if we manage to reduce epistemic uncertainty, e.g., via prompting or few-shot examples, the model may become capable of rejecting samples of high aleatoric uncertainty.
>
> Our additional experiment results (Appendix G) verify the feasibility:
> Instead of directly asking the VLM to perform galaxy classification, we can prompt the model to answer questions in Galaxy Zoo's decision tree (https://data.galaxyzoo.org/gz_trees/gz_trees.html),
> which decomposes galaxy classification into a sequence of simple visual questions with no domain knowledge of astrophysics needed.
> This can be viewed as augmenting the model verbally with the knowledge of *how to recognize galaxy images*.
>
> As a proof of concept, we considered the first question in the decision tree (Smooth v.s. Featured galaxy), which decides whether the image falls into `(smooth_round, smooth_cigar)` v.s. `(edge_on_disk, unbarred_spiral)` categories.
> Broadly, given carefully designed prompts, where we explain the concepts of the categories and rejection conditions in plain language
> - On instances where human annotators reach a consensus, the model's answer aligns with the annotators' consensus (Figure 10)
> - On instances where human annotators show high disagreement, the model rejects the instance by returning ``Unknown'' (Figure 11).
>
> Note that we do find that the model's rejection behavior is more sensitive to the prompting design in this case compared with the ImageNet-C cases.
> We believe the prompt sensitivity can be reduced through fine-tuning to internalize the knowledge and conditions for rejection.
>
>
> # Question-conditional ambiguity (Weakness 1 & 2)
>
> Yes, we agree that caption diversity does not capture the question-conditional ambiguity.
>
> Our intuition for caption diversity is built upon a generative classifier P(X, y | query) = P(y | X, query)P(X), where caption diversity captures the variance in the question-independent visual features X.
>
> This highlights the advantage of caption diversity:
> It is not bound to any specific type of query.
> But also reflects its limitation, it does not take into account the question-conditional uncertainty as you suggested.
>
> However, evaluating and solving question-conditional uncertainty goes beyond the scope of our work.
> We agree that this is a very interesting and challenging research question that arises uniquely in the multi-modality setting.
>
> # Comparison between caption diversity and logit-based (Weakness 3, Requested changes 3)
>
> First of all, caption diversity, as you suggested, is not related to the query; therefore, it does need a specific question (e.g. object classification), in contrast to logit-based.
>
> However, if we do have a query, we do not expect caption diversity to show a significant advantage (performance or efficiency-wise) over logit-based, or other language modality-only uncertainty quantification approaches, such as semantic entropy.
> Therefore, in the manuscript, we are not claiming caption diversity as a metric for uncertainty quantification, but more as a tool for understanding and diagnosing the underlying mechanism of how VLM perceives input uncertainty.

---

> > ### Author Response · Authors · 2026-01-05
> > **Author response (2/2)**
> >
> > # Prompting strategies vs. coverage (weakness 4)
> >
> > This is a great point!
> > We believe that different prompts will certainly result in different accuracy-rejection trade off, similiar to the effect of adjusting the threshold in classification with rejection.
> > However, it is unclear how to adjust the trade-off through tweaking the prompt, which is one limitation of our proposed prompting-to-reject method.
> >
> > We believe this motivates another valuable future direction:
> > Prompting to rejection is easy to use and efficient to implement.
> > However, **how can we adjust the level of rejection using prompting?**
> > A potential solution is to include a special verbal flag in the prompt, similar to how the reasoning effort is controlled verbally in the prompt for GPT-OSS.
> > The verbal flag (e.g. `aggressive / mild / none`) will adjust the rejection level, and the model can be post-trained, using a specific accuracy-rejection trade-off as the target for reward, to understand and use such a flag.

---

### Review · Reviewer_V15z · 2025-12-21

**Summary Of Contributions:**

This paper evaluates the robustness of modern vision-language models (VLMs) against two classic uncertainty quantification challenges: anomaly detection and classification under ambiguous inputs. Its core contributions include: (1) Empirically demonstrating that newer/larger VLMs outperform earlier models in handling corrupted/OOD inputs but still tend to hallucinate confident responses without proper prompting; (2) Proposing a simple yet effective solution—explicitly prompting models to abstain from uncertain predictions— which achieves near-perfect reliability on standard benchmarks (e.g., ImageNet-C, CIFAR-10 vs. Non-CIFAR-10) without architectural modifications; (3) Introducing caption diversity as a label-free metric to reveal VLMs’ internal uncertainty, enabling prediction of model rejection behavior.

Key strengths: Rigorous experimental design with diverse datasets (natural images, ECG signals, galaxy morphology) and model families; practical, low-cost intervention (prompting) with clear real-world utility; novel insight linking caption diversity to uncertainty.

Key weaknesses: Limited solution for domain-specific tasks (e.g., galaxy classification) where VLMs fail due to insufficient expertise; insufficient analysis of why certain models (e.g., Llama 3.2) are more sensitive to prompting styles; lack of comparison with state-of-the-art uncertainty quantification methods for VLMs.

**Audience:**

Yes

**Audience Explanation:**

TMLR’s audience includes researchers focused on machine learning, deep learning robustness, and multimodal models—groups for whom this work is highly relevant. Uncertainty quantification is a critical topic for safe deployment of AI systems, and VLMs are increasingly adopted in real-world applications (e.g., medical imaging, autonomous systems). The paper addresses a pressing gap: while VLMs are scaled to massive sizes, their ability to handle ambiguous/anomalous inputs remains understudied. The findings (e.g., prompting as a quick fix, caption diversity for uncertainty estimation) provide actionable insights for practitioners and open new directions for research on VLMs’ reliability. Additionally, the contrast between general and domain-specific task performance informs future work on domain-adapted VLMs, making the paper of interest to both fundamental and applied researchers.

**Broader Impact Concerns:**

N/A.

**Claims And Evidence:**

Yes

**Claims Explanation:**

The paper’s claims are strongly supported by comprehensive empirical evidence. First, the authors design well-controlled experiments across multiple tasks (anomaly detection, ambiguous classification) and datasets, covering both general and domain-specific scenarios. Second, they benchmark 6 VLMs (spanning different sizes and release dates) and report quantitative metrics (precision, recall, F1-score, accuracy, rejection rate) alongside qualitative examples (e.g., Figure 2a’s noisy image classification). Third, visualizations (e.g., Figure 5 on caption diversity vs. corruption level) and ablation studies (Appendix E on prompting styles) validate causal links between prompting, model behavior, and uncertainty. All experiments are described in detail (e.g., dataset construction, sampling parameters) to ensure reproducibility, aligning with TMLR’s evaluation criteria for accurate and convincing evidence.

**Requested Changes:**

- Address the limitation in domain-specific tasks: Provide concrete proposals (e.g., integrating domain knowledge via prompt engineering, few-shot fine-tuning, or hybrid models) to improve VLMs’ uncertainty quantification in specialized fields like galaxy morphology. This gap currently undermines the paper’s generalizability.
- Expand analysis of prompting style sensitivity: Explain why older models (e.g., Llama 3.2) rely on “caption & answer” prompts while newer models (e.g., Qwen 2.5 72B) are robust to prompt variations. This analysis will strengthen the understanding of how VLMs process uncertainty instructions.
- Compare with existing uncertainty methods for VLMs: Include a brief comparison with state-of-the-art techniques (e.g., Bayesian VLMs, ensemble-based methods) to contextualize the proposed prompting approach’s advantages (e.g., efficiency, simplicity) and limitations.
Extend experiments to additional VLMs (e.g., Gemini Pro, Claude 3) to confirm the generalizability of the prompting and caption diversity findings.
- Supplement Appendix E’s ablation study with qualitative examples of how different prompts alter model reasoning (e.g., verbatim outputs for “direct” vs. “caption & answer” prompts).
- Discuss practical trade-offs of the rejection prompt (e.g., false rejection rates in high-stakes scenarios) and provide guidelines for choosing prompt styles based on model capabilities.
- Clarify the mechanism behind caption diversity: Include a short discussion on why ambiguous inputs lead to more diverse captions (e.g., conflicting visual features, lack of domain knowledge) to deepen the theoretical grounding.

---

> ### Author Response · Authors · 2026-01-05
> **Author response (1/2)**
>
> We would like to thank the reviewer for the detailed review.
>
> Please see below for our detailed response to the items in Requested Changes.
>
> # Solution for Galaxy Zoo
>
> As you suggested, we believe the solution lies in augmenting the VLM with domain knowledge.
>
> One promising solution for Galaxy Zoo is to prompt the model to follow the questions in the decision tree (https://data.galaxyzoo.org/gz_trees/gz_trees.html) rather than directly performing classification. The decision tree is composed of questions designed by astrophysics experts to help human annotators without domain knowledge perform galaxy morphological classification.
>
> Our additional experiment results (Appendix G) provide a proof of concept. We considered the first question in the decision tree (Smooth v.s. Featured galaxy), which decides whether the image falls into `(smooth_round, smooth_cigar)` v.s. `(edge_on_disk, unbarred_spiral)` categories.
> Broadly, under carefully designed prompts, where we explain the concepts of the categories and rejection conditions in plain language
> - On instances where human annotators reach a consensus, the model's answer aligns with the annotators' consensus (Figure 10)
> - On instances where human annotators show high disagreement, the model rejects the instance by returning ``Unknown'' (Figure 11).
>
> Note that we do find that the model's rejection behavior is more sensitive to the prompting design in this case compared with the ImageNet-C cases.
> We believe the prompt sensitivity can be reduced through fine-tuning to internalize the knowledge and conditions for rejection.
>
> The main idea here is that we could think of out-of-box VLM as a generic human annotator without domain knowledge, equipped with basic visual recognition ability.
> Then, with the assistance of experts, we can decompose the domain-specific tasks into a series of basic visual subtasks and prompt the VLM to finish them, where the models are better at quantifying uncertainty.
>
>
> # Why certain models (e.g., Llama 3.2) are more sensitive to prompting styles;
>
> In short, newer models may be instructional-finetuned to answer "I don't know".
>
> To be more specific, newer models could have an improved visual instruction tuning dataset, with questions where the ground truth answer is "Unknown / I don't know".
> Note that it is already common practice to include "I don't know" type of question in LLM pre-training [1]
> But considering that multi-modality LLM is a fairly new area, where the visual instruction tuning dataset is one of the most important ingredients for training [1], the lab may not include these special instructions in early revisions, since this is more of an additional feature.
>
> Note that if we do caption and answer, then there are more verbal information in the context (the caption), therefore VLM can utilize the rejection ability from the base LLM to perform rejection.
>
> [1] https://www.wsj.com/articles/google-and-anthropic-are-selling-generative-ai-to-businesses-even-as-they-address-its-shortcomings-ff90d83d
>
> [2] https://arxiv.org/abs/2304.08485
>
> # Comparison of prompting with other uncertainty quantification methods for VLMs.
>
> Thanks for raising this very good point.
>
> Bayesian VLM (probabilistic CLIP, [1]), Bayesian / Ensembled LoRA ([2, 3]), are not applicable to out-of-box VLM as they require architecture modifications or further fine-tuning, in contrast to prompting, which works with any instruction-tuned VLMs.
>
> Sampling-based approaches, such as semantic entropy [4], require generating multiple answers, making it less efficient than rejection prompting, which requires only one generation.
>
> Logit-based approaches directly look at the probability of each options and use the entropy or maximum softmax probability as measurement for uncertainty. Logit-based approaches require specially structured prompts (e.g. multiple-choice) and access to the generation probabilities (which may not be available for certain closed-source model). In contrast, the prompting-based approach relies only on the output texts and no internal information.
>
> The major limitation of prompting compared with sampling-based and logit-based is that we can't adjust the level of rejection by changing a threshold.
> For these two methods, we reject the sample when the uncertainty score exceeds a certain threshold value, and the threshold balances between the rejection rate and accuracy. When using prompting, it is solely the model's decision to choose how aggressively it wants to reject, and we believe studying how adjusting the implicit rejection threshold through prompting could be a promising future direction.
>
>
> [1] https://proceedings.mlr.press/v286/venkataramanan25a.html
>
> [2] https://arxiv.org/abs/2308.13111
>
> [3] https://arxiv.org/abs/2310.00035
>
> [4] https://arxiv.org/abs/2302.09664

---

> > ### Author Response · Authors · 2026-01-05
> > **Author response (2/2)**
> >
> > # How different prompts alter model reasoning
> >
> > Thanks for the suggestion!
> >
> > We have updated Appendix E with qualitative examples to illustrate the model's reasoning under "simple", "direct" and "caption & answer". In particular, we present the sampled outputs from Llama3.2, with high sensitivity to prompting, and outputs from Qwen2.5 7B, which show little variation to prompting, under a corruption level 5 image.
> >
> > # Discuss practical trade-offs of the rejection prompt (e.g., false rejection rates in high-stakes scenarios) and provide guidelines for choosing prompt styles based on model capabilities.
> >
> > Enabling rejection in prompting will certainly introduce the risk of false negatives, i.e. rejection by mistake.
> > However, in high-stakes scenarios, it is often safer for the model to behave conservatively, since the rejected samples can always be deferred to human experts for the final decision, whereas failing to reject ambiguous samples could introduce high cost in downstream decision-making problems.
> >
> > We believe for models newer than the ones we considered in the paper, i.e. those released after Feb 2025, prompting styles won't have a significant impact, as newer models tend to have better instruction following ability.
> >
> > # Mechanism behind caption diversity
> >
> > Yes, we believe, as you suggested, the mechanism behind diverse captions lies in ambiguity in the visual features, which results in multiple ways to interpret the contents.
> >
> > However, the model must first have the domain knowledge to recognize the image in order to understand the ambiguity and generate diverse captions.
> > Our results indicate that on ambiguous galaxy images, VLMs fail to generate diverse captions because they don't know how to read the images.

---

### Author Response · Authors · 2026-01-28

Dear editor, dear reviewers

We would like to thank you again for your time and efforts for providing such high quality feedbacks.

We would appreciate it if we could receive further feedback or a decision soon, since it affects whether we can submit this to the upcoming conference deadline.

At the same time, we are also open to further questions and feedbacks and would love to provide clarifications and adjustments.

Best,

Authors

---

### Decision · Action_Editor_TbWy · 2026-02-07

**Recommendation:** Accept with minor revision

**Additional Comments:**

The following minor edits are necessary:

1. Revise the abstract and introduction to specify which VLMs were tested and acknowledge that reasoning-focused models were not evaluated. Frame findings in terms of the tested models rather than implying universality.
2. Section 3.2 should explicitly frame caption diversity as an empirical diagnostic proxy here and not a general uncertainty estimator (the exposition leaves this question underspecified: where are not Bayesian here etc etc). Acknowledge that diverse captions can arise from multiple sources (visual ambiguity, rich scene content) and discuss applicability limitations more prominently.
3. The rebuttal's clarification re epistemic and aleatoric uncertainty should appear in the main paper and appendix to hypothesize on the Galaxy Zoo performance. Also briefly mention the Appendix G decision-tree experiments as a proof-of-concept for domain-knowledge augmentation and/or sufficient task simplification and first evidence of the epistemic uncertainty explanation.
4. Include a paragraph in the related work to compare rejection prompting against Bayesian VLMs, ensembles, semantic entropy, and logit-based methods, as provided in the rebuttal. A few sentences on tradeoffs (efficiency vs. controllability, access requirements) might also be good.

These revisions should be straightforward as this has been covered by the authors’ existing responses and will be verified before the camera-ready is accepted

**Audience:**

Yes

**Audience Explanation:**

The paper addresses an important practical question about VLMs. The suggested approaches are simple to implement and have clear practical value for anyone deploying VLMs and can inspire further research questions. The additional results in Appendix G show that more specific prompting and simpler tasks (or task decomposition) could help where the models lack domain expertise, which points towards further potential research.

**Claims And Evidence:**

Yes

**Claims Explanation:**

The paper's claims are supported by its experiments across six VLMs (GPT-4o-mini, Llama 3.2 11B, Qwen 2 7B, Qwen 2.5 3B/7B/72B) and multiple datasets (CIFAR-10C, ImageNet-C, CIFAR-10 vs. non-CIFAR-10, ECG, Galaxy Zoo 2). The main paper and appendices cover the methodology and dataset construction details. Additional ablation studies on prompting strategies are also provided. The paper and author response clearly acknowledge and detail limitations.

One reviewer provided specific concerns about caption diversity and about claim scope relative to experimental evidence, which are addressable through a minor paper revision.

Another reviewer changed their "Claims And Evidence" assessment to "No" in their final recommendations but, despite repeated direct inquiry, did not provide a rationale for this.

Given the authors addressed and committed to the requested changes in their rebuttal, I find the evidence criterion is met after having read the paper in detail myself.

---

> ### Author Response · Authors · 2026-02-18
> **Camera ready version**
>
> Dear Action Editor,
>
> Thank you very much for your careful reading of our paper and monitoring the review process. We truly appreciate the time and effort you invested in synthesizing the reviews and clearly outlining the necessary revisions.
>
> Please see below for a summarization of the changes in the revised PDF per your comment
>
> ## Scope of tested model
>
> We have revised the abstract and the intro to be more explicit on the scope of the tested models:  "...Our results based on non-reasoning VLMs released between late 2024 and early 2025..."
>
> ## Caption diversity
> In section 3.2, we have made the following changes
> - Rephrase caption diversity as a tool to diagnose VLM's understanding of **input ambiguity** rather than an uncertainty measurement.
> - Included an additional paragraph ("Scope and limitation of caption diversity") explicitly stating that caption diversity is not a generic  uncertainty estimator as well as some reasons beyond corruption that could trigger high caption diversity.
>
> ## Aleatoric, epistemic uncertainty and galaxy zoo
>
> In the end of Sec 4.2, we include an additional paragraph "Hypothesis and potential solution for GalaxyZoo's failure", which includes our clarification during the rebuttal as well as discussion to the results in Appendix G.
>
> ## Additional related work
>
> In the last sentence of first paragraph, Sec 2.2, we discussed the simplicity and low access requirements for rejection prompting as well as the cost: Limited controllability.
>
> We additionally include a paragraph in Sec 2.2 comparing ensemble / Bayesian VLM with prompting.

---

> > ### Author Response · Authors · 2026-03-06
> > **Author follow up**
> >
> > Dear editor
> >
> > We am writing to follow up on our camera-ready revision and to ask whether the revised version is satisfactory from your perspective.
> >
> > We have carefully addressed the requested changes and submitted the updated manuscript accordingly. We would be very grateful if you could let us know whether the revision meets the requirements for the camera-ready version, or if there are any additional issues that still need our attention.
> >
> > Thank you very much for your time and help throughout this process.
> >
> > Thanks,
> >
> > Authors

---

> > > ### Comment · Action_Editor_TbWy · 2026-03-07
> > > **Two edits requested:**
> > >
> > > Two typos:
> > >
> > > "ambiguous score" -> "ambiguity score"?
> > >
> > > p 11: "; For Galaxy Zoo" -> ". For Galaxy zoo"
> > >
> > > Looks great otherwise.
> > >
> > > Thanks! Please fix and re-upload and I can certify.
> > >
> > > Sorry for the delay.

---

> > > > ### Author Response · Authors · 2026-03-07
> > > > **Edits done!**
> > > >
> > > > Thanks for catching the typos! They are fixed in the latest revision.
> > > >
> > > > Thanks,
> > > >
> > > > Authors